# Lignocellulose-Degrading Enzymes: A Biotechnology Platform for Ferulic Acid Production from Agro-Industrial Side Streams

**DOI:** 10.3390/foods10123056

**Published:** 2021-12-08

**Authors:** Vitalijs Radenkovs, Karina Juhnevica-Radenkova, Jorens Kviesis, Danija Lazdina, Anda Valdovska, Fernando Vallejo, Gunars Lacis

**Affiliations:** 1Processing and Biochemistry Department, Institute of Horticulture, Graudu Str. 1, LV-3701 Dobele, Latvia; karina.juhnevica-radenkova@llu.lv (K.J.-R.); danija.lazdina@llu.lv (D.L.); 2Research Laboratory of Biotechnology, Division of Smart Technologies, Latvia University of Life Sciences and Technologies, Rigas Str. 22, LV-3004 Jelgava, Latvia; gunars.lacis@llu.lv; 3Department of Environmental Science, University of Latvia, Jelgavas Str. 1, LV-1004 Riga, Latvia; jorens.kviesis@lu.lv; 4Research Laboratory of Biotechnology, Division of Agronomic Analysis, Latvia University of Life Sciences and Technologies, Strazdu Str. 1, LV-3002 Jelgava, Latvia; Anda.Valdovska@llu.lv; 5Faculty of Veterinary Medicine, Latvia University of Life Sciences and Technologies, Kr. Helmana Str. 8, LV-3004 Jelgava, Latvia; 6Metabolomics Service, CEBAS-CSIC, University Campus of Espinardo, Edif. 25, 30100 Murcia, Spain; fvallejo@cebas.csic.es; 7Unit of Genetics and Breeding, Institute of Horticulture, Graudu Str. 1, LV-3701 Dobele, Latvia

**Keywords:** biorefining, bran, enzymatic hydrolysis, green extraction, hydroxycinnamates, sustainability, valorization

## Abstract

Biorefining by enzymatic hydrolysis (EH) of lignocellulosic waste material due to low costs and affordability has received enormous interest amongst scientists as a potential strategy suitable for the production of bioactive ingredients and chemicals. In this study, a sustainable and eco-friendly approach to extracting bound ferulic acid (FA) was demonstrated using single-step EH by a mixture of lignocellulose-degrading enzymes. For comparative purposes of the efficiency of EH, an online extraction and analysis technique using supercritical fluid extraction–supercritical fluid chromatography–mass spectrometry (SFE-SFC-MS) was performed. The experimental results demonstrated up to 369.3 mg 100 g^−1^ FA release from rye bran after 48 h EH with Viscozyme L. The EH of wheat and oat bran with Viscoferm for 48 h resulted in 255.1 and 33.5 mg 100 g^−1^ of FA, respectively. The release of FA from bran matrix using supercritical fluid extraction with carbon dioxide and ethanol as a co-solvent (SFE-CO_2_-EtOH) delivered up to 464.3 mg 100 g^−1^ of FA, though the extractability varied depending on the parameters used. The 10-fold and 30-fold scale-up experiments confirmed the applicability of EH as a bioprocessing method valid for the industrial scale. The highest yield of FA in both scale-up experiments was obtained from rye bran after 48 h of EH with Viscozyme L. In purified extracts, the absence of xylose, arabinose, and glucose as the final degradation products of lignocellulose was proven by high-performance liquid chromatography with refractive index detection (HPLC-RID). Up to 94.0% purity of FA was achieved by solid-phase extraction (SPE) using the polymeric reversed-phase Strata X column and 50% EtOH as the eluent.

## 1. Introduction

Cereals make a significant contribution to the economy of the EU, and their by-products are thought to be a potential renewable source of high-demand bioactive compounds. According to the Food and Agriculture Organization Corporate Statistical Database (FAOSTAT, 2021) [1], the global cereal production quantity, including wheat, sorghum, rye, rice, oats, millet, maize, and barley (average), over the past 50 years increased by 240.5% from 108.3 million tons (Mt) in 1961 to 368.9 Mt in 2019 [1]. Due to their availability, the by-products are attracting enormous attention amongst researchers worldwide as a potential and renewable raw material suitable for the manufacturing of active biomolecules and added-value functional ingredients [2]. However, due to a lack of innovations in processing of grain-derived by-products and existing limitations in effective full-fledged transfer of data, grain by-products still are used as feedstuff. The presence of indigestible dietary fiber in the matrix of bran makes this material suitable for livestock feeding, while their application in the food industry is negligible. Valorization of by-products could be done through the application of innovative and green strategies aimed at biomass transformation by extracellularly produced hydrolases, such as cellulases, xylanases, and feruloyl esterases.

It has been demonstrated that wheat bran could be used as a raw material for the synthesis of fumaric acid by enzymatic hydrolysis (EH) of acid-pretreated wheat bran with *Rhizopus oryzae* [3]. Bioconversion of wheat-bran-derived ferulic acid (FA) through the use of a mixture of commercial enzymes and an *Escherichia coli* JM109 (pBB1) strain could deliver up to 90.0 mg L^−1^ of vanillin, a flavoring compound that is highly demanded and widely used in the food industry [4]. Within the EH of grain-derived by-products, selective production of FA could be achieved by cellulolytic and xylanolytic enzymes that are capable of depolymerizing β-d-(1→4)-glucosidic and β-d-(1→4)-xylosidic bonds of cellulose and hemicellulose polymers, respectively [5]. More recently, Ferri et al. [6] proposed a sequential treatment of thermally pretreated wheat bran for the selective release of FA through the application of a blend of proteolytic, amylolytic, and cellulolytic enzymes. The authors highlighted the efficiency of the developed technology, since a 40-fold scale-up EH process provided a satisfactory yield of FA. These results were further confirmed by Martín-Diana et al. [7], demonstrating the ability of the multi-active β-glucanase and β-xylanase enzyme complex Ultraflo XL to release FA from wheat bran under optimal conditions.

Rye bran is less explored; however, like wheat, it can be biorefined and further used as a source of such bioactives as benzoxazinoids, phenolic acids, and alkylresorcinols [8]. This assertion was already reinforced by Kapreliants and Zhurlova [9], indicating that biomodification of rye bran within stepwise EH of thermally and mechanically pretreated bran material aids in biotechnologically obtaining high-value bioactives, i.e., leucine, arginine, and valine. Similarly to wheat, the hydroxycinnamates (HCMs) in rye bran (FA specifically) are present as components of the outer layer of the kernel, which are located in the outer and inner pericarps, seed coats (testa), hyaline, and aleurone layers, and are tightly attached to hemicellulose through ester bonds [10]. The ratio of free-to-bound FA in the rye arabinose-xylose (hemicellulose) fraction is 0.1:1000, indicating a strong binding affinity of this molecule to other macronutrients. The report of Konopka et al. (2014) [11] revealed the presence of 0.0082 mg 100 g^−1^ free FA in whole rye grain flour, pointing to the relative abundance of this phenolic acid in rye grain with other cinnamates. Alkaline-assisted hydrolysis with subsequent solid–liquid or liquid–liquid extraction seems to be a feasible solution to release bound forms. The report of Andreasen et al. [12] indicates the presence of FA in the range of 90.0–117.0 mg 100 g^−1^ in 17 analyzed alkaline-hydrolyzed rye varieties. Inconsistent results, however, were reported by Buksa et al. [13], showing that the content of FA after 24 h alkaline-assisted hydrolysis of a fraction rich in arabinoxylans was found to be considerably lower than that of non-treated samples. The authors pointed out that harsh conditions during hydrolysis had led to the quantitative and qualitative loss of phenolic acids, including FA. A similar observation was made by [14], indicating the formation of 4-vinylguaiacol as the first degradation product of FA that appeared during alkaline hydrolysis.

Although the release of FA using alkaline-assisted hydrolysis of bran material is high, unluckily its production under industrial scale requires a substantial input of strong alkalis and acids. However, regarding already existing greenhouse gas emission problems, the use of strong alkalis or acids would negatively influence human health, the environment, and industry. In a recent study by Juhnevica-Radenkova et al. [14], up to 135.6% of the total alkaline-extractable FA release was achieved from rye bran after EH with a sole multi-enzyme complex Viscozyme L. However, this is the only report highlighting the release of FA during EH with lignocellulose-degrading enzymes from rye bran available so far.

Oats, due to their nutritional composition and availability of multiple bioactive compounds, such as β-glucan, avenanthramides, tocopherols and tocotrienols, phytosterols, phytic acid, and avenacosides, are reported to have beneficial effects on human health, including a reduction in cardiovascular diseases, type 2 diabetes mellitus, gastrointestinal disorders, and cancer risk [15]. It has been stated that among phenolic compounds, FA is found to be the dominant HCM in oats; however, similarly to wheat and rye, FA is integrated into the matrix of the three-dimensional structure of cellulose, hemicellulose, and lignin, forming strong ester or ether linkages with it [16]. The report of Zhao and Moghadasian [17] reveals that whole oats may contain 25–35 mg 100 g^−1^ of FA, while oat bran may contain 33.0 mg 100 g^−1^; however, growing conditions and variety are the main factors determining the amount of FA.

In the context of the European Green Deal process, EH eliminates the use of toxic and/or corrosive chemicals and, due to the relative simplicity of operational conditions, could represent the future strategy for the production of compounds that are currently manufactured by chemical routes.

The limited information about the recovery of FA and other HCMs from rye, wheat, and oat bran using green processing technologies promoted the design of this study, focusing on the evaluation of the release of FA from bran as a result of biorefining accomplished by three lignocellulose-degrading enzymes.

## 2. Materials and Methods

### 2.1. Plant Material

Three types of commercial food-grade bran samples were obtained from a local supplier Voldemars Ltd. (Riga, Latvia) separated as rye (*Secale cereále* L.), wheat (*Triticum aestívum* L.), and oats (*Avena sativa* L). Based on morphological evaluation, hydrological layers such as the outer pericarp, inner pericarp (cross cells, tube cells), seed coat (testa), hyaline, and aleurone layers with attached starch granules were identified in the bran samples. The proximate composition of the bran samples is shown in Table 1.

### 2.2. Plant Material Preparation for Alkaline and Enzymatic Hydrolysis and Analysis of Hydroxycinnamates

Preparation of bran material for further work and analysis was carried out following the methodology described by Radenkovs et al. [18], while the moisture content of bran was analyzed gravimetrically, as proposed by Ruiz [19].

### 2.3. Chemicals and Reagents

Commercial standards, i.e., caffeic acid (CA), *trans*-isomer of ferulic acid (*t*-FA) and *trans*-isomer of iso-FA (*t*-iso-FA), vanillic acid (VA), vanillin (VN), *p*-coumaric acid (*p*-CA), *trans*-cinnamic acid (*t*-CA), (−)-epicatechin, (±)-catechin, gallic acid (GA), sinapic acid (SA), syringic acid (SIA), protocatechuic acid (PCA), 4-vinylphenol (4-VP), 2-methoxy-4-vinylphenol (4-VG), eugenol (EGN), neochlorogenic (NCGA) and chlorogenic acids (CGA), rhamnose, ribose, xylose, arabinose, sorbose, fructose, mannose, glucose, galactose, sucrose, maltose, lactose, and glycerol, were purchased from Sigma-Aldrich Chemie Ltd., (Steinheim, Germany). A standard solution containing a mixture of C_4_-C_24_ fatty acid methyl esters (FAMEs) with purity ≥99.0% were acquired from Sigma-Aldrich Chemie Ltd. Sodium hydroxide (NaOH), potassium hydroxide (KOH), citric acid (C_6_H_8_O_7_), 2,2-diphenyl-1-picrylhydrazyl, sodium citrate dihydrate (C_6_H_5_Na_3_O_7_·2H_2_O), phenolphthalein (C_20_H_14_O_4_), and 0.5M trimethylphenylammonium hydroxide solution (CH_3_)_3_N(OH)C_6_H_5_ (TMPAH) in methanol (MeOH) for gas chromatography (GC) derivatization were of reagent grade. Ethanol (EtOH), MeOH, acetonitrile (MeCN), formic acid (HCOOH) (puriss r.a.), and ammonium formate (HCO_2_NH_4_) of liquid chromatography-mass spectrometry (LC-MS) grade were purchased from Merck KGaA (Darmstadt, Germany). HPLC-grade diethyl ether (C_2_H_5_), *n*-hexane (C_6_H_14_), and pyridine (C_5_H_5_N) were purchased from Merck KGaA. Ultrapure water was produced using the reverse osmosis PureLab Flex Elga water purification system (Veolia Water Technologies, Paris, France).

### 2.4. Enzymes

Commercial lignocellulose-degrading enzymes were supplied in kind by the company Novozymes^®^ (Bagsvaerd, Denmark) for laboratory purposes. Since each preparation selected is a multi-enzyme complex containing various cellulolytic and xylanolytic enzymes, in this study they were applied independently, not as a mixture. A list of enzymes applied is depicted in Table 2.

### 2.5. Enzymatic Hydrolysis of Rye, Wheat, and Oat Bran

EH of bran samples using biocatalysts was performed in an SW23 water bath with a capacity of 20.0 L and a horizontal shaking system (Julabo^®^, Zalbaha-Hinterglemm, Germany). To assess the impact of the duration of EH on the release of HCMs (FA specifically), the process of EH was carried out over the range of 12–72 h. The optimal operational conditions for each enzyme were chosen individually based on the Novozymes^®^ (Bagsvaerd, Denmark) recommendations and supporting the protocol described by Juhnevica-Radenkova et al. [14]. The EH of lignocellulose, i.e., hemicellulose and cellulose, was performed using three commercially available multi-enzyme complexes, i.e., Viscozyme L, Celluclast 1.5 L, and Viscoferm. For this purpose, 10 mL of 0.5 M sodium citrate buffer (pH 4.6) containing 6 FBG mL^−1^ of endo-1,4-β-xylanase (Viscozyme L or Viscoferm) or 10 EGU mL^−1^ of endo-1,4-β-d-glucanase (Celluclas 1.5 L) was added to 1 g of rye, wheat, and oat bran samples. The obtained blends were then vortex-mixed for 2 min using the ZX3 vortex mixer (Velp^®^ Scientifica, Usmate Velate, Italy), followed by incubation in a water bath at 44 ± 1 °C and 100 rpm. The release of HCMs at the above-mentioned time points of EH was estimated chromatographically. To complete the reaction, before HPLC analysis, each aliquot harvested was subjected to ultrasonication at 50 kHz with an output wattage of 360 W for 10 min at 25 ± 1 °C using an Ultrasons ultrasonic bath (J.P. Selecta^®^, Barcelona, Spain) and centrifugation at 20,160× *g* for 10 min at 25 ± 1 °C in a Hermle Z 36 HK centrifuge (Hermle Labortechnik, GmbH, Wehingen, Germany). Filtration was done using a 0.22 µm polyvinylidene fluoride (PVDF) hydrophilic membrane filter (Durapore, Millipore, Billerica, MA, USA).

### 2.6. Hydrolysis of Bran Samples under the 10- and 30-Fold Scale-Up Process

For FA release from the bran matrix, scale-up experiments were performed using the New Brunswick Bioflo/Celligen 115 bioreactor (Eppendorf, Hamburg, Germany). The process was carried out in a 2.5 L glass vessel at a scale of 10 and 30 g of bran. The same ratio of buffer to substrate was used. During EH, the prepared mixture was continuously stirred at 100 rpm using the Rushton impeller. The pH of the mixture was maintained at 4.6 by the proportional and integral controller that operates peristaltic pumps, assigned to perform acid or base addition. The measurements of pH were performed using a gel-filled pH probe (Eppendorf, Hamburg, Germany). The release of HCMs at different time points, i.e., 12, 24, 48, and 72 h, of EH was ascertained chromatographically within 2h. The obtained hydrolysates were ultrasonicated at 50 kHz for 10 min at 25 ± 1 °C and filtered through a 0.22 µm PVDF before chromatographic analysis.

### 2.7. Recovery of FA from Bran Hydrolysates by Solid-Phase Extraction

FA was extracted from rye bran hydrolysates following a protocol provided by Phenomenex with minor modifications. Briefly, 3 mL of bran hydrolysate was spiked with an internal standard (3,5-dichloro-4-hydroxybenzoic acid), followed by filtration through a 0.45 µm PVDF membrane filter. Recovery of FA was done using solid-phase extraction (SPE) with a Strata-X column (Phenomenex, Torrance, CA, USA) filled with a styrene-divinylbenzene-based reversed-phase polymer (33 µm, 85 Å, 30 mg 3 mL^−1^). Conditioning/equilibration of the SPE column was done using 1 column volume of pure MeOH, followed by 1 column volume of DDW. The loaded sample was washed with 1 column volume of DDW, and a flow-through fraction was collected for further chromatographic work and analysis. Analyte elution was done using 2 volumes of either acidified absolute MeOH or EtOH (2% formic acid *v/v*) or their aqueous solutions. Collected eluate fractions were subjected to drying under a gentle stream of N_2_ to complete dryness. The dried samples were reconstituted in 1 mL of 80% acidified MeOH (MeOH:DDW:formic acid ratio 80:19:1 *v**/v/v*).

### 2.8. Plant Material Preparation for SFE-SFC Extraction of Hydroxycinnamates

Duplicate samples of 100 ± 1.0 mg of dried and ground (Ø ≤ 0.5 mm particle size) bran material were weighed into 5 mL stainless steel extraction cells (Shimadzu Corporation, Tokyo, Japan) with cellulose discs at the bottom and top. The extraction vessel was placed in the rack changer of the SFE apparatus. Liquid CO_2_ and EtOH as a co-solvent (modifier) were delivered through pumps into the extraction vessel and changed to supercritical fluid by adjusting the temperature and pressure.

Variations in extraction conditions used in the experiments to assess the release of HCMs are depicted in Table 3, while other parameters are as follows: the extraction solvent consisted of solvent A (supercritical fluid of CO_2_ (AGA, Latvia; purity > 99.0)) and solvent B (modifier; EtOH, purity 96.8%) delivered to the extraction vessel at a flow rate of 5.4 mL min^−1^ using back pressure regulator A (14.7 MPa) and back pressure regulator B (15.0 MPa), both operated at 50 °C.

### 2.9. The Online SFE-SFC Extraction Conditions for Hydroxycinnamates

Along with alkaline hydrolysis, the protocol described in detail by Juhnevica-Radenkova et al. [14], the online extraction, separation, and analysis of HCMs using supercritical fluid extraction–supercritical fluid chromatography (SFE-SFE) coupled to a triple quadrupole (TQ) mass-selective detector (MS-8050) was used in this experiment.

The online supercritical fluid extraction–supercritical fluid chromatography–mass spectrometry (SFE-SFC-TQ-MS) extraction and analysis was performed using the Nexera UC SFE-SFC-LC system (Shimadzu Corporation, Tokyo, Japan) with the following configuration: SFE-30A (SFE module), LC-30ADSF (CO_2_ deliver pump), LC-40DX3 (modifier deliver pump), DGU-405 (degassing unit), CTO-40AC (column oven), SFC-30A × 2 (back pressure adjustment module), Rack Changer II, TQ-MS-8050 (mass-selective detector), SCL-40 (system controller), and LabSolutions Insight LC-MS version 3.7 SP3 (workstation) (Shimadzu Corporation, Tokyo, Japan), as shown in Figure 1.

### 2.10. The Online SFE-SFC-TQ-MS/MS Analytical Conditions for Hydroxycinnamates

Chromatographic separation of HCMs was carried out using a reversed-phase Shim-pack UC-RP column (5.0 μm, 250 × 4.6 mm; Tokyo, Japan) operating at 45 °C and a flow rate of 1.0 mL min^−1^. The mobile phase used was a supercritical fluid of CO_2_ (A) and acidified MeOH (0.002 % formic acid, *v/v*) with 5 mM ammonium formate (B). Separation of HCMs was done using the following gradient conditions: elution started with 10% B to obtain 40% B at 11 min, 60% B at 15 min, and 10% B at 17 min, with a subsequent 2 min wash and equilibration. A make-up solution consisting of 5 mM ammonium formate in MeOH was delivered with a make-up pump after column separation to promote HCM ionization. Data were acquired using LabSolutions Insight software, which was also used for instrument control and processing. The ionization in both positive and negative ion polarity modes was applied in this study, while data were collected in profile and centroid modes, with a data storage threshold of 5000 absorbance for MS. The operating conditions were as follows: detector voltage 1.8 kV, conversion dynode voltage 10.0 kV, interface voltage 4.0 kV, interface temperature 300 °C, desolvation line temperature 250 °C, heat block temperature 400 °C, nebulizing gas argon (Ar, purity 99.9%,) at flow 3.0 L min^−1^, heating gas carbon dioxide (CO_2_, purity 99.0%,) at flow10.0 L min^−1^, and drying gas nitrogen (N_2_, separated from air using a nitrogen generator system from Peak Scientific Instruments Ltd. (Inchinnan, Scotland, UK), purity 99.0%) at flow10.0 L min^−1^. All HCMs were observed in the programmed and optimized multiple reaction monitoring (MRM) mode. All MRM transitions, collision energy, Q1, Q3, and dwell time for phenolic compounds are depicted in Table 4.

### 2.11. The HPLC-ESI-TQ-MS/MS Analytical Conditions for Phenolics

The analyses were carried out using a Shimadzu series Nexera UC SFC-SFE-LC system (Tokyo, Japan) coupled to TQ-MS-8050 (Tokyo, Japan) with an electrospray ionization interface (ESI). A sample of 1 μL was injected onto a reversed-phase Shim-pack UC-RP column (5.0 μm, 250 × 4.6 mm; Tokyo, Japan) operating at 45 °C and a flow rate of 1.0 mL min^−1^. The mobile phases used were acidified DDW (0.002% formic acid, *v/v*) supplemented with 5 mM ammonium formate (A) and acidified MeOH with 5 mM ammonium formate (0.002% formic acid, *v/v*) (B). Separation of compounds was done using the following gradient conditions: elution started with 5% B to obtain 10% B at 5 min, 60% B at 12–15 min, and 10% B at 18 min. Furthermore, MeOH injections were included every three samples as a blank run to avoid the carry-over effect. Data acquisition, analysis, and processing were performed similarly as for the online SFE-SFC-TQ-MS/MS analytical conditions for hydroxycinnamates (Section 2.10).

### 2.12. The HPLC-RID Analytical Conditions for Carbohydrates

The quantitative analysis of mono- and disaccharides, hydrolysates, and analytical standards was performed on a Waters Alliance HPLC system (model No. e2695) equipped with a 2414 RI detector and a 2998 column heater (Waters Corporation, Milford, MA, USA). Chromatographic separation was performed on an Altima Amino (4.6 × 250 mm; 5 μm; Grace™, Columbia, MD, USA) column. The column and flow cell temperature was maintained at 30 °C. A mixture of DDW and CH_3_CN (80:20, *v/v*) was used as the mobile phase in isocratic mode. The flow rate of the mobile phase was 1.0 mL min^−1^. The injection volume was 15 μL. System control, data acquisition, analysis, and processing were performed using Empower 3 Chromatography Data Software version (build 3471) (Waters Corporation, Milford, MA, USA).

### 2.13. Preparation of the Lipid Fraction by Alkaline-Assisted Hydrolysis and Liquid–Liquid Extraction

For hydrolysis of the bran matrix and release of the bound forms of fatty acids, 10% (*w/v*) KOH dissolved in 80% MeOH (MeOH:DWW ratio 80:20 *v/v*) was used. In excess of MeOH, this method allows the process of hydrolysis and release of fatty acids to be performed more efficiently. Triplicate samples of 5 ± 0.1 g of ground bran were weighed in 25 mL reagent bottles with screw caps. For the hydrolysis, 20 mL of prepared methanolic KOH was added to bran and the mixture was subjected to incubation in a TW8 water bath (Julabo^®^, Saalbach-Hinterglemm, Germany) at 65 °C for 3 h. After hydrolysis, the release of fatty acids from the salt form was performed by shifting the pH of the medium from alkaline to acidic by adding 6 M HCl until the pH was 2.0. The extraction of the lipophilic fraction was accomplished by liquid–liquid phase separation using *n*-hexane as a solvent. Briefly, 10 mL of *n*-hexane was added to the prepared hydrolysates (total 15 mL), followed by vortex mixing for 1 min and separation of the layers by centrifugation at 4500 rpm (3169× *g*) for 10 min in a Sigma, 2-16KC centrifuge (Osterode near Harz, Germany). The top *n*-hexane layer was separated and collected. The extraction procedure was repeated three times. The resulting lipophilic fraction (30 mL) was further evaporated using a Laborota 4002 rotary evaporator (Heidolph, Swabia, Germany) at 65 °C, and the dry fraction was then re-dissolved in 2 mL of pyridine and filtered through a polytetrafluoroethylene (PTFE) hydrophobic membrane filter with a pore size 0.45 µm (VWR™, International, GmbH., Darmstadt, Germany) The filtrates were quantitatively transferred to 22 mL glass headspace chromatography bottles for further chromatographic work and analysis.

### 2.14. Preparation of Fatty Acids for GC/MS Analysis

TMPAH reagent was used as a methylating agent of the polyfunctional groups to obtain volatile fatty acid derivatives. The methylation procedure was performed according to the protocol described in the American Society for Testing and Materials [20]. Briefly, a 5 µL aliquot of the lipid fraction was taken from the separated lipophilic fraction and 5 µL of 1% phenolphthalein indicator (C_20_H_14_O_4_:EtOH ratio 1:99 *w/v*), 6 µL of 0.5 M TMPAH reagent, and 930 µL of methanol:diethyl ether (MeOH:Et_2_O ratio 50:50 *v/v*) were added to the vial. The resulting pink mixture was vortex-mixed for 1 min and kept in a GC oven at 60 °C for 30 min. The mixture was than cooled to room temperature and further used for FAME analysis on a GC/MS system. Quantification of compounds was performed using a standard solution of C_4_-C_24_ FAMEs, building calibration curves for each compound individually.

### 2.15. The GC Conditions for FAME Analysis

The analysis of fatty acid methyl esters (FAMEs) was performed using a Clarus 600 system (PerkinElmer, Inc., Waltham, MA, USA) coupled to a single quadrupole Clarus 600 C mass-selective detector (Waltham, MA, USA). The chromatographic separation of FAMEs was done using a Trace™ TR-FAME (Thermo Fisher Scientific, Waltham, MA USA) column with a cyanopropylphenyl-based stationary phase (50 m × 0.22 mm, sorbent thickness 0.25 μm) specifically designed for the separation of *cis*- and *trans*-isomers of FAMEs (Appendix A). The injector temperature was set to +280 °C; automatic injection was performed using an autosampler at an injection volume of 0.5 μL and a split ratio of 4:1. The initial oven temperature was kept at 70 °C for 2 min, then increased to 150 °C (rate of 20 °C min^−1^), and then raised to 250 °C (rate of 4 °C min^−1^). Helium (ultra-high-purity 5.0 grade 99.999%) was used as a carrier gas at a constant flow rate of 1.0 min with a split ratio of n:1. The total separation time was 31 min. The analysis was performed in triplicate.

### 2.16. The MS Conditions for FAME Detection

The detector mode was an electron ionization system with ionization energy 70 eV, ion source temperature +230 °C, MS transfer line temperature +280 °C, capture time starting from 6.5 min (1.7 scan s^−1^), ion multiplier 240 V, and ion *m*/*z* interval 41–500 atom mass units (AMU) for FAMEs.

### 2.17. Statistical Analysis

The results obtained are shown as means ± standard deviation of the mean from three replicates (*n* = 3). A *p*-value of <0.05 was used to denote significant differences between mean values determined using one-way analysis of variance (ANOVA) and Duncan’s multiple-range test performed using IBM^®^ SPSS^®^ Statistics version 20.0 (SPSS Inc., Chicago, IL, USA).

## 3. Results and Discussion

### 3.1. Release of FA from Bran Using Enzyme-Assisted Hydrolysis

Since nearly 97% of FA in cereal bran is bound with various macromolecules, forming cross-links, the efficient release/extraction of this compound presents a challenge. The alkaline hydrolysis of complex agro-industrial by-products was reported as an effective tool for the extraction of FA and other HCMs. However, concerning environmental pollution matters and governmental intentions outlined in the EC Directive 2010/75/EU [21], aiming to reduce the negative impact of industrial toxic emissions on ecosystems creates an additional demand for developing green technologies for the production of chemicals from renewable sources.

To establish the ability of lignocellulose-degrading enzymes to release bound FA forms from bran matrices, including rye, wheat, and oats, EH with a single process step was used in this study. In this set of experiments, bran samples underwent 12–72 h EH solely by lignocellulose-degrading enzymes, i.e., Viscozyme L, Celluclast 1.5 L, and Viscoferm. The highest yield of *t*-FA in all cases of enzymes applied was obtained from rye and wheat bran after 48 h of EH, except for wheat bran and Viscozyme L, where the highest yield was acquired already after 24 h (Figure 2A,B).

However, upon EH, a significant loss of *t*-FA by 16.5–37.2% and 18.7–99.4% was observed for rye and wheat bran, respectively, indicating a further degradation process caused by thermal decarboxylation of HCMs, as revealed by Ohra-Aho et al. [22]. This observation is reinforced by the presence of 2-methoxy-4-vinylphenol (4-VG) (4-vinyl guaiacol) and 4-allyl-2-methoxyphenol (eugenol), the degradation products of FA and Klason lignin, respectively (Figure 3A,B).

Among the hydrolytic enzymes examined, the advantage of Viscozyme L over other enzymes was noticed during the processing of rye bran. This observation was also established by Mahmoudi et al. [23], working with the enzyme-assisted extraction of bioactives from sweet basil by-products. However, conducting EH of wheat bran with the multi-enzyme complex Viscoferm, up to 1.4-fold higher yield of *t*-FA was obtained compared with EH with Viscozyme L. It is worth noting, though, that the selective MRM-MS-based approach revealed no presence of *t*-FA in wheat bran hydrolysates that underwent EH with Viscozyme L for 48 h. The absence of *t*-FA in hydrolysates can be explained by the ability of the *trans*-isomer of FA to be isomerized to the *cis*- form under UV irradiation that, perhaps, took place during the next 24 h of EH [24]. Notably, a smaller release of *t*-FA was found in oat bran (Figure 2C), but along with *t*-FA, an equivalent yield of other HCMs was detected in obtained hydrolysates. The concentration of *t*-FA was almost 11.7-fold and 7.9-fold lower than that released from rye and wheat bran, respectively. A similar amount of *t*-FA released after alkaline-assisted hydrolysis of oat bran was reported by [25].

The second prevalent HCM isolated from bran hydrolysates and further successfully quantified was *t*-iso-FA. The concentration of *t*-iso-FA in bran hydrolysates fluctuated from 1.26 to 182.4 mg 100 g^−1^ (on a dry weight basis), depending on the enzyme and the duration of EH used. Among the three hydrolytic enzymes screened, the highest yield of *t*-iso-FA, similarly to *t*-FA, was obtained in rye bran samples subjected to 48 h EH with the multi-enzyme complex Viscozyme L. A considerably smaller release of *t*-iso-FA from the rye bran matrix was observed after 48 h EH with Viscoferm, though the amount was 1.3-fold higher than with Celluclast 1.5 L. The presence of *t*-iso-FA in the grains and grain-derived fractions of such crops as wheat, oats, barley, corn, red rice, and rye has already been confirmed by [26,27,28], indicating that rye contains the highest amount, but the ratio of free-to-bound *t*-iso-FA is 10:90. However, the amounts described herein are 4.4- and 28.4-fold higher than those previously reported for wheat and oat bran, respectively.

The most available literature on clinical trials refers to *t*-iso-FA derived from *Cimicifugae rhizoma*, a medical herb with multiple well-documented health-promoting benefits [29,30], although little attention has been given to other sources of *t*-iso-FA. However, despite the beneficial health effects outlined in the report of Li and Yu [31], there is little generalized information about the exact concentration of *t*-iso-FA. It was hypothesized that the grain-derived hydrolysates discussed herein may have the same beneficial health effects and could be used as a remedy in the prevention and treatment of various ailments, which is also supported by [28].

Sinapic acid (SA) is the third HCM the presence of which was confirmed in all bran hydrolysates obtained by selected enzymes. The advantage of using Viscoferm over other enzymes was noticed during the processing of wheat bran, since this preparation was able to deliver up to 18.75 mg 100 g^−1^ of SA within 24 and 48 h of EH. The yield of SA in this study is analogous to that obtained by alkaline-assisted hydrolysis of whole wheat grains [32]. The authors highlighted that SA after FA and dimeric FA is the third-most prevalent HCM released from whole wheat grains. However, contradicting results were observed in the case of rye bran, where the yield of SA was highest in the sample that underwent EH with Viscozyme L for 24 and 48 h. This time, a comparable amount of SA was found in oat bran hydrolysates after EH with Celluclast 1.5 L, and the highest yield was observed after 48 h of EH. The EH of oat bran with Viscoferm and Viscozyme L regardless of the EH duration delivered the lowest yield of SA. Similar to this study, the advantage of a developed biorefining process aimed at releasing SA from mustard bran was demonstrated in the report of Achinivu et al. [33], highlighting better release of SA than that from alkali-processed bran.

The small-scale process revealed that each enzyme contributes differently to the release of HCMs from the bran matrix due to its specificity and activity, though among the enzymes tested, the superiority of the multi-enzyme complex Viscozyme L must be highlighted. Overall, a gentler approach ensured up to 215.7% release of the total alkali-extractable FA from rye bran, while 79.5% and 60.3% release could be achieved from wheat and oat, respectively.

### 3.2. Release of FA from Bran Using Enzyme-Assisted Hydrolysis of the 10- and 30-Fold Scale-Up Process

The EH at high solid loading is another important key to the scale-up process for the release of FA from complex matrices. Due to the presence of higher free sugars and other bioactives that can act as inhibitors and therefore interfere with the release of target compounds, the process of scaling is necessary to establish the applicability of the developed technology. The increase in solid loading could significantly reduce the efficiency of mass and heat transference, influence the degree of substrate homogeneity, and limit the formation of the enzyme–substrate complex by restricting the access of the enzyme to the available substrate, resulting in lower solubilization of cellulose, hemicellulose, β-glucan, and other non-soluble dietary fiber. There have hitherto been developed various pretreatment strategies regarding the productivity, yield, and number of steps to enhance the hydrolysis of lignocellulose materials. Over the past few years, pilot-scale operational conditions have been proposed for alkaline [34,35], hydrothermal [36], and ultra-fast hydrolysis by supercritical CO_2_ pretreatments. More recently, Ferri et al. [6] demonstrated a robust and eco-friendly three-step biorefinery process for the efficient release of FA. The proposed method seems to be valid for the industrial-scale production of FA. So far, however, there is limited information about large-scale EH of rye bran, especially at high solid loading, so the current experiment will be focused on scaling up the EH process accomplished by three hydrolytic enzymes in a single step.

A 10- and 30-fold scale-up was carried out using a bioreactor and parameters developed for small-scale FA release. The release of FA and other HCMs was analyzed after 48 h of EH, since reduction in the content of both FA isomers was observed for wheat bran, while no significant changes in the release of HCMs took place for rye bran during further 24 h of a small-scale EH process (Figure 4). Besides, additional energy input would make this process not economically feasible, resulting in a higher price of the final product.

A 10- and 30-fold scale-up process confirmed the applicability of EH as a bioprocessing type valid for industrial-scale FA production. Similarly, as in the small scale, the advantage of Viscozyme L over other hydrolytic enzymes has been proved in scaling-up the process (Figure 4A).

Extracted ion chromatograms (EICs) representing HCMs detected in the rye-bran-derived hydrolysate are depicted in Figure 5.

At 10- and 30-fold scales, up to 431.9 and 372.6 mg of FA (sum of *t*-FA and *t*-iso-FA) can be produced from 100 g of rye bran by the way of EH, respectively. However, as can be seen, the highest concentration of FA was reached at the 10-fold scale-up experiment. It is worth noting that the amount of FA released at the bioreactor scale was found to be significantly higher than was observed at smaller scales. A plausible explanation for obtaining a better release of FA was provided by Ferri et al. [6], who posited that bioreactor scale conditions improve EH performance as a result of controlled pH and continuous agitation of the slurry. All this made it attainable to reach a more homogeneous enzymatic reaction and hydrolysis efficiency.

As with small-scale EH, the better release of FA from wheat bran was observed during EH with Viscoferm (Figure 4B). The produced amount of FA in both 10- and 30-fold scale-up experiments was significantly higher than observed for other enzymes. Using the proposed conditions, 100 g of EH wheat bran could deliver up to 263.6 and 250.2 mg of FA, respectively. As seen, a significantly higher yield of *t*-FA and *t*-iso-FA was found for the 10-fold scale-up experiment.

Due to gentler hydrolysis conditions, the scaled-up EH process of rye and wheat bran with Viscozyme and Viscoferm demonstrated up to 217.6% and 91.2% release of the total alkali-extractable FA (sum of *t*-FA and *t*-iso-FA) from rye and wheat bran, respectively.

### 3.3. Release of FA from Bran Using SFE-CO_2_ Extraction

The interactions between intracellular phytochemicals and cell walls have a significant influence on the extractability of compounds of interest from complex plant matrices [5], including whole grains and their derived fractions. Recently, a couple of dozen extractants were investigated and proposed for the industrial-scale production of dietary polyphenols, generally favoring solvents or their blends, such as acetone [37], methanol [38], diethyl ether, and ethyl acetate [39], even though they may be harmful to operators and the environment. Among the extraction methods documented, superior extractability of phytochemicals from complex matrices was reported for SFE [40]. Due to the GRAS status assigned and relatively low costs, CO_2_ is the most widely used supercritical fluid suitable for both research purposes and industrial scales. Since neat supercritical fluid CO_2_ has dissolving properties close to those of hexane, which is recognized as an excellent solvent for extracting non-polar compounds, the addition of co-solvent could enhance the solubilizing properties of CO_2_, making it attainable to recover more polar molecules. Due to the process taking place in a closed loop, the online extraction, separation, and analysis of HCMs using SFE-CO_2_-SFC coupled to TQ-MS/MS make it attainable to reduce both qualitative and quantitative losses of analytes during analytical work. Given this circumstance, along with reduced solvent consumption, the SFE technique has a significant advantage over the other sample preparation methods [41].

For comparative purposes of the EH efficiency, the SFE-CO_2_ extraction coupled to SFC-TQ-MS/MS was applied in this experiment. Since extraction parameters or their interactions, such as temperature, pressure, co-solvent concentration, and mode (static and dynamic), could influence the yield compounds of interest, a series of SFE-SFE operational conditions were investigated in a lab-scale Nexera UC SFE-SFC-LC-MS-QT apparatus (Table 3).

In the first test runs 1–4, the influence of extraction temperature over the range of 40–60 °C was investigated, while pressure, co-solvent concentration (modifier), and extraction parameters were kept constant at 10 MPa, 10% EtOH, and 3:3 min, respectively (Figure 6). The selected temperature range was based on a previous report [42]. As expected, the yield of FA and other HCMs increased with the increase in extraction temperature, while with the temperature rising over 50 °C, the amount of *t*-FA decreased significantly (*p* < 0.05) and the observed value was even lower than that obtained at 45 °C (Figure 6A). The results are consistent with those of [43,44], indicating that the vast majority of phenolics and antioxidants present in grape seeds and cranberry bush berries and pomace are extracted using a temperature range of 44–46 °C. A plausible explanation for obtaining a better release of FA was given by Ameer et al. [40], pointing out that elevated temperature after the plant matrix is wetted allows reducing the surface tension and increasing solvent cavitation that accelerates the dissolution of analyte in the solvent.

It is worth noting that the presence of *t*-iso-FA was found only when rye bran samples were subjected to extraction at 50 °C. Therefore, an extraction temperature of 50 °C was used for subsequent extractions. The third prevalent compound extracted from rye bran samples by means of SFE-CO_2_ was VA. The yield of this HCM fluctuated in the range of 133.2–166.4 mg 100 g^−1^. It should be admitted that the value of VA was found to be significantly higher than that yielded during EH, indicating a possible decarboxylation process of free VA caused by elevated temperature during hydrolysate autoclaving. The presence of 4-VG as the primary degradation product of VA was also confirmed by [45].

To evaluate the effect of pressure on the extractability of FA from the rye bran matrix by SFE-CO_2_, experiments (5–8) were carried out using extraction pressure within the range of 10–25 MPa at a constant temperature (50 °C), concentration of modifier (10%), and length of static to dynamic mode (3:3 min). As seen (Figure 6B), the yield of *t*-FA was significantly higher when rye bran samples were subjected to 25 and 15 MPa extraction pressures. However, when extracting at 25 MPa, the presence of *t*-iso-FA was not detected, revealing higher susceptibility of *t*-iso-FA to degradation by elevated pressure than of *t*-FA. However, when calculating the total amount of these two HCMs, the obvious advantage of using 15 MPa pressure among all other pressures is noticed. Considering the yield of other HCMs, it is seen that there were no significant differences in the content of CA, *p*-CA, and VN; therefore, the pressure of 15 MPa was selected for further extractions to minimize wear of the machine parts.

To study the influence of co-solvent concentration on the extractability of FA from rye bran, experiments in the next series of runs (9–12) were conducted. The addition of EtOH in the range of 7.5–20% along with the main supercritical CO_2_ extraction fluid was used in this experiment, while moderate operating conditions such as temperature 50 °C, pressure 15 MPa, and the length of static to dynamic mode 3:3 min were maintained constant (Figure 6C). Since *t*-FA and *t*-iso-FA present both methoxy and hydroxy substituents, and also to keep the SFE-CO_2_ extraction green, polar EtOH was selected as a co-solvent. The results showed that almost equal yields of *t*-FA could be reached with the addition of polar EtOH to the CO_2_ extraction fluid in the range of 7.5–10%, though the increase in the extraction solvent polarity and elevated viscosity resulted in a marked reduction in *t*-FA solubility and recovery. The yield of *t*-iso-FA in the fraction obtained with the use of 10% EtOH as a co-solvent was found to be the highest. In turn, no presence of *t*-iso-FA was found in the extracts obtained with the addition of 15% and 20% EtOH. The results are consistent with data obtained in an earlier study [46], highlighting that due to the relatively low toxicity and potential applications of obtained extracts in food, cosmetic, and pharmaceutical products, this solvent has advantages over others. The total FA content was relatively higher by performing the extraction with the addition of 10% EtOH; therefore, this concentration was selected for subsequent extractions.

In the final set of experiments (13–17), when optimal values for temperature, pressure, and concentration of co-solvent for extracting FA from the bran matrix were elucidated, the influence of extraction mode, i.e., static and dynamic, and its length was investigated. As it turned out, the type of extraction has played a crucial role that could affect the yield of FA (Figure 6D). The lowest yield of *t*-FA was obtained when the extraction was performed in 5 min static and 1 min dynamic mode; however, the amount of *t*-iso-FA in the extracts was the highest. To find a compromise between the yield of both *t*-FA and *t*-iso-FA, the extraction of HCMs was performed using a 4 min static and 2 min dynamic mode. As can be seen, an 89.9% higher yield of *t*-FA was reached, while no presence of *t*-iso-FA was detected in the extracts obtained using this mode. A satisfactory yield of *t*-FA and *t*-iso-FA was reached only after shifting from 5 min static extraction mode to 5 min dynamic mode (static-to-dynamic ratio 1:5). A credible explanation for obtaining better extractability of compounds applying a longer dynamic mode was provided by Luque de Castro et al. [47], pointing out that continuous exposure of the analyte to the clean extraction solvent could enhance displacement of the analyte, partitioning equilibrium to the mobile phase.

To summarize, the introduced SFE-SFC approach is another eco-friendly alternative suitable for industrial-scale production of FA. The extractability of FA (sum of *t*-FA and *t*-iso-FA) using CO_2_ and 10% EtOH as a co-solvent, 15 MPa pressure, 50 °C temperature, a static mode length of 1 min, and a dynamic mode length of 5 min delivered an increase in the obtained FA by 164.0% in rye, 69.9% in wheat, and 329.9% in oat bran in comparison to FA obtained after alkaline-assisted hydrolysis (Figure 6E). It is worth noting that technologies involving elevated pressures require high investment costs for high-pressure equipment, and therefore the application of SFE must be justified by the extraction of substances, which, along with added value, are also in demand on the market and are of potential industrial application [48]

### 3.4. Release of Mono- and Disaccharides from Bran Using Enzyme-Assisted Hydrolysis

According to Novozymes^®^, hydrolytic enzymes selected in this study contain a broad spectrum of carbohydrases. Under proper conditions, depending on the substrate, such end products as glucose, arabinose, mannose, galactose, and xylose, can be released as a result of their synergistic hydrolytic activity. It has been demonstrated that FA both in grains and in grain-derived fractions is presented in a covalently cross-linked form with arabinofuranosidase residues in arabinoxylans, while continuous EH with lignocellulose-degrading enzymes results in solubilization of water-insoluble arabinoxylans and depolymerization of water-soluble arabinoxylans with the simultaneous release of FA [5]. This statement was reinforced by Shin et al. [49], revealing a strong correlation between the release of FA and the yield of reducing sugars during EH of corn fiber.

Previous experiments have revealed the ability of selected hydrolytic enzymes to release bound FA forms from the rye, wheat, and oat bran matrix within a single process step of EH. Therefore, the next experiment will be aimed at assessing the yield of mono- and disaccharides over 72 h of EH. A detailed profile of individual sugars after small-scale EH is depicted in Table 5. Glucose, xylose, arabinose, and fructose were the major products released from rye, wheat, and oat bran (Appendix A), though the amount of sugars varied depending on the hydrolytic enzyme and duration of EH. The presence of galactose was also confirmed in bran hydrolysates, though further work with direct enzyme injections revealed the availability of this sugar as part of commercial enzyme preparations.

The action of α-L-arabinofuranosidase resulted in the cleavage of the non-reducing end of terminal α-L-arabinofuranoside residues in α-L-arabinosides, and therefore, the presence of feruloylated arabinose monomers at the initial stage of EH was revealed. Viscoferm displayed the maximum hydrolytic performance after 12 h of EH since the highest yield of arabinose in rye bran hydrolysates was 8.0 g 1000 mL^−1^. The obtained results are consistent with data obtained by Gama et al. [50], indicating that arabinofuranosidase is more active at the initial stage of EH, since by the action of this enzyme, smaller xylan polysaccharides are formed that contain no side chains of arabinose residues. The action of this enzyme ensures the xylan area is more accessible to other enzymes involved in splitting of xylosidic bonds with the simultaneous release of xylooligosaccharides along with xylose monomers. However, it is worth noting that at the end of 72 h, a 28.4% decrease in this monosaccharide was noticed. A similar trend of arabinose decrease was found in rye bran hydrolysates subjected to EH with Viscozyme L, where up to 40.6% of arabinose loss was observed after 72 h of EH. Opposite results were obtained during EH of wheat bran samples with Viscoferm, where up to a 150.5% increase in arabinose was observed at the end of 72 h. Considerable fluctuations in this monomer content were detected during EH of wheat bran with Viscozyme L, the maximum peak of which was marked upon 48 h of EH. However, additional 24 h hydrolysis resulted in a 44.4% reduction in this monomer. A credible explanation of this phenomenon was given by Xin et al. [51], pointing out that continuation of the hydrolytic activity of α-L-arabinofuranosidase can cause extensive loss of free arabinose.

Xylose is a sugar monomer that can be released during EH through the action of xylanolytic enzymes that are capable of depolymerizing xylosidic bonds present in the xylan backbone chain. The highest content of xylose monomers was observed after 48 h of EH with Viscozyme L. Up to 12.8 and 10.6 g 1000 mL^−1^ of xylose was released from rye and wheat bran samples, demonstrating superior hydrolytic activity of this enzyme over other applied enzymes. Contrary results were reported by Bautista-Expósito et al. [52], demonstrating the superior xylanolytic performance of Viscoferm, since the amount of xylose released was 7.4-fold higher than when using Viscozyme L.

According to Novozymes^®^, the main activity of hydrolytic enzymes chosen is being glycosidic; in addition to the aldopentose (arabinose and xylose) content, the efficiency of hydrolysis was evaluated by measuring the amount of aldohexose (glucose) in hydrolysates. The results of glucose content showed values ranging between 5.9 and 62.3 g 1000 mL^−1^ for rye, wheat, and oat bran hydrolysates, with oat bran having the highest content and wheat bran the lowest. A higher amount of glucose in oat bran hydrolysates is explained by the presence of the relatively higher amount of β-glucan, which alongside cellulose releases glucose monomers by the action of endo-1,4-β-d-glucanase [53]. The maximum yield of glucose was obtained after 48 h of EH with Viscozyme L. The same observation was made by Bautista-Expósito et al. [52], highlighting that Viscozyme L over 13 other commercial enzymes tested is the most efficient, able to release the highest amount of glucose monomers from wheat bran polysaccharides. EH with Viscozyme L for 48 h led to up to 76.5%, 81.5%, and 5.1% increase in glucose content as compared to the initial level obtained after 12 h of rye, wheat, and oat bran EH, respectively. The highest release of glucose using Viscoferm was reached only passing 72 h of EH, corresponding to an 83.7% and 62.9% increase for rye and oat bran hydrolysates, respectively, while continuous EH of wheat bran samples during 72 h failed to release higher amounts of glucose.

### 3.5. Release of Mono- and Disaccharides from Bran Using Enzyme-Assisted Hydrolysis of the 10- and 30-Fold Scale-Up Process

A 10- and 30-fold scale-up process was carried out using a bioreactor. The release of individual sugars was analyzed after 48 h of EH, since during further 24 h of the small-scale process due to the continuation of the hydrolytic activity of α-L-arabinofuranosidase present in Viscozyme L, an extensive reduction in arabinose content was observed in all bran hydrolysates. A scale-up process revealed a similar trend of individual sugars increasing after 48 h of EH. The advantage of Viscoferm over other hydrolytic enzymes was proven in both scale-up experiments at high solid loading, though the amount released varied depending on the type of sugar analyzed. As was seen, the amount of arabinose in bran hydrolysates was found in the range from 1.2 to 9.0 and from 0.9 to 7.1 g 1000 mL^−1^ for 10- and 30-fold scale-up experiments, respectively. The highest arabinose content was observed in rye bran hydrolysates that underwent EH with Viscoferm and Viscozyme L, though it was 97.2% and 37.7% higher than that observed during small-scale EH, respectively (Figure 7A,D). As shown, the highest concentration of xylose monomer in both scale-up experiments was reached after EH with Celluclast L, with rye bran having the highest content (Figure 7A,D) and oat bran the lowest (Figure 7C,F).

The amount of xylose released was 18.6% (10-fold) and 25.5% (30-fold) lower than that observed with small-scale rye bran EH. Viscozyme L demonstrated the lowest hydrolytic activity at high solid loading concerning xylose yield, which is possibly associated with the high susceptibility of xylanase to external factors, such as high content of other sugars monomers or availability of free bioactives that exhibit inhibitory activities. However, assessing the glucose yield obtained in the scaled-up EH process, the superiority of Viscozyme L over other enzymes was highlighted for rye and wheat bran in comparison with the small-scale EH experiment. The amounts of glucose obtained under both 10- and 30-fold scale-up conditions were 7.8–8.8%, 19.0–38.3%, and 8.0–9.1% lower than those released during small-scale EH of rye, wheat, and oat, respectively.

Evaluation of the total amount of sugars revealed lower release in both 10- and 30-fold scale-up experiments than that acquired during the small-scale EH process; however, the proposed EH conditions are still valid, since 100 g of enzymatically hydrolyzed rye, wheat, and oat could deliver up to 55.3, 32.5, and 60.0 g of sugars, respectively. In general, the highest total sugar yield was obtained during EH of bran with Viscozyme L, except for wheat (10-fold scale), where Viscoferm demonstrated relatively better hydrolytic performance. In the context of non-waste technology, the mono- and disaccharides released have great potential to be used in bioethanol fuel production; however, a proper purification process needs to be developed [34,54].

### 3.6. Recovery of FA from Bran Hydrolysates

In this experiment, the hydrolysates of rye bran acquired after EH with Viscozyme L were subjected to SPE purification using a polymeric reversed-phase Strata X column (30 mg 3 mL^−1^). The selection of the sorbent was based on availability, effectiveness, and simplicity of use, since no specific activation, i.e., thermal, acidic, or base, aside from conditioning with either MeOH or EtOH is required [55].

The solubility of FA can vary depending on the solvent and the concentration used. In the earlier report of Couteau et al. [56], positive desorption of FA from polyvinylpolypyrrolidone (PVPP) with a purity of about 62% was achieved by 96% EtOH, while later, 97.4% recovery of FA was demonstrated by 60% EtOH [57]. The contradiction in results on the recovery of FA using different concentrations of alcohol promoted the design of this experiment to study the recovery and purity of FA using MeOH and the less toxic EtOH and their aqueous solutions as elution solvents. The purity of FA in collected fractions was studied chromatographically by assessing the presence of side HCMs and mono- and disaccharides.

The results on FA recovery using MeOH and its aqueous solutions revealed increasing desorption of both *t*-FA and *t*-iso-FA in a MeOH-concentration-dependent manner (Figure 8A). Up to 67.2% and 80.1% recovery of *t*-FA and *t*-iso-FA from rye bran hydrolysates was achieved through acidified 80% MeOH (2% formic acid, *v/v*), respectively. The overall purity of FA in the fraction collected was 82.7%. Application of acidified 90% MeOH caused a rise in the recovery of *t*-FA and *t*-iso-FA, since 70.6% and 83.6% of these HCMs were observed in the collected phase, but the relative purity of FA was 83.4%. To improve the purity of FA, an additional wash-up step with 5% MeOH was performed after FA-rich hydrolysates were passed through the column and afterward eluted by 90% MeOH. This approach brought up the purity of FA to 90.1% with a relative recovery of 71.6%.

Satisfactory results on the recovery of FA from rye bran hydrolysates were obtained with the application of EtOH as an eluent (Figure 8B). At the first step of purification, 20% and 40% EtOH was used to check the validity of these solvents for desorbing of FA retained on the stationary phase of the Strata X column. As revealed, the selected concentrations of EtOH were found be to less effective in desorbing either *t*-FA or *t*-iso-FA from the polymer selected. However, the recovery of these two compounds was 67.5% and 97.5% more effective than that achieved by MeOH, respectively. An increase in the EtOH concentration up to 50% resulted in a rise in the recovery of *t*-FA and *t*-iso-FA of up to 99.1% and 92.9%. Application of this eluting agent ensured 94.0% purity of FA obtained.

### 3.7. Fatty Acid Composition of Brown Lipophilic Substance

Buranov et al. [58] indicated in a report that the alkaline-derived hydrolysates of wheat and corn bran contain a brown lipophilic substance that makes the process of FA purification challenging. The presence of the brown substance was also mentioned in a report by Salgado et al. [55], noting that it can be easily precipitated by adding 30% EtOH to the alkali extract, followed by centrifugation. However, neither of the reports specified the composition of the waxy substance observed; therefore, in a further section, characterization of the lipophilic fraction will be performed based on the fatty acid (FTA) content (Table 6).

The results demonstrated significant variations in FTA percentage distribution between hydrolysis methods used. The abundance of unsaturated FTA was revealed for lipids obtained after EH of bran, making up 77–81% of the total FTA (in relative percentage), while 43–61% of unsaturated FTA was observed in lipids of alkaline-assisted hydrolysates. Linoleic acid (C18:2), oleic acid (C18:1), and palmitic acid (C16:0) contributed the most to the total amount of FTA, corresponding to about 93.0% to 98.0%. The results on the relative percentage of FTA in rye and wheat bran are consistent with data obtained by Cardoso et al. [59], noting that C18:2, followed by C18:1, C16:0, and, to a lesser extent, C18:3, is the most abundant FTA found. The amount of SFA in the lipid fraction of oat bran varied in the range of 4.1–9.6%, while PUFA was in the range of 39.4–46.3%. C18:1 was found to be the most dominant fatty acid, the percentage of which was in the range of 13.1–29.0%. A similar contribution of SFA to the total amount of FTA was reported for oat lipids after alkaline hydrolysis, followed by methylation in boron trifluoride-methanol [60]. A higher amount of PUFA was found in rye-derived lipids upon subjecting bran samples to EH with Celluclast 1.5 L and alkaline-assisted hydrolysis, since the concentration was 16.0–23.7% and 18.7% higher than that observed in oat-bran-derived lipids, respectively.

The total content of FTA in bran-derived lipids is hydrolysis method dependent and varied in the range of 21.4 ± 1.0 to 101.4 ± 6.0 g 100 mL^−1^. The advantage of Celluclast L over other enzymes used concerning FTA release from wheat bran samples was observed. The amount of FTA in wheat-derived lipids was found to be 289.9% higher than that obtained by alkaline hydrolysis. Notably weaker but still effective release of FTA from the rye and wheat bran matrix was observed upon EH with Viscozyme L. EH of rye and wheat bran in comparison to alkaline-assisted hydrolysis ensured the release of 68.4 ± 7.8 and 51.3. ± 2.4 g 100 mL^−1^ of FTA, which is 160.9% and 87.8% more than what alkaline hydrolysis does. EH with Viscoferm in comparison to alkaline hydrolysis was able to release 25.4% and 18.8% more FTA from the oat and wheat bran matrix, while no obvious effect was observed for rye bran.

In general, considering Viscozyme L and Viscoferm have higher declared xylanolytic than cellulolytic activity, which was also confirmed by Wikiera et al. [61], it can be assumed that the degradation of cellulose-composed cell wall membranes using cellulolytic enzymes is more important for the efficient release of FTA than using xylanolytic enzymes.

## 4. Conclusions

The main intention of this study was to elucidate the hydrolytic ability of lignocellulose-degrading enzymes and release bound ferulic acid (FA) during enzymatic hydrolysis (EH) of bran. For these purposes, three types of bran, i.e., rye, wheat, and oats, were subjected to EH. The release of mono- and disaccharides, FA, and other hydroxycinnamates (HCMs) was monitored throughout the entire process of EH. For comparative purposes, the efficiency of EH in releasing FA, along with alkaline-assisted hydrolysis, was assessed by an online SFE-SFC-MS extraction and analysis technique. The process of small-scale EH revealed that each enzyme, due to its specificity and activity, contributes differently to the release of HCMs from the bran matrix, though among the enzymes tested, the advantage of Viscozyme L was highlighted. Overall, a gentler approach demonstrated up to 215.7% release of the total alkali-extractable FA (sum of *t*-FA and *t*-iso-FA) from rye bran, while 79.5% and 60.3% was released from wheat and oat bran, respectively. Favorable EH conditions during the scale-up process ensured FA release from rye and wheat bran comparable to the small-scale process. Up to 217.6% and 91.2% release of the total alkali-extractable FA was achieved from rye and wheat bran when performing EH with Viscozyme L and Viscoferm, respectively. The introduced SFE-CO_2_-EtOH extraction approach revealed a significantly higher release of FA, though the extractability varied depending on the parameters used. Extraction by CO_2_ and 10% EtOH as a co-solvent, 15 MPa pressure, 50 °C temperature, and the length of the static and dynamic mode of 1 and 5 min was able to deliver a 164.0%, 69.9%, and 329.9% higher amount of FA than what was obtained after alkaline-assisted hydrolysis of rye, wheat, and oat bran, respectively. The experimental results on recovery and purification of FA demonstrated the applicability of the styrene-divinylbenzene-based reversed-phase Strata X column for removal of the degradation products from bran hydrolysates. In the purified fraction, the absence of residual HCMs and carbohydrates was confirmed, obtaining recovered FA of 94.0% purity. Compared to the SFE technique, EH could be considered an affordable alternative, which can represent the future of sustainable FA production, while the side products, mono- and disaccharides, could be exploited as renewable raw material for bioethanol fuel production or, along with remaining solids, might be introduced in the feed formulations for cattle.

In the context of non-waste technology, our proposed method has the potential to be used under large-scale conditions, where industrial symbiosis by combining the grain milling facility as a supplier of bran and biorefinery for the production of FA can be demonstrated.

## Figures and Tables

**Figure 1 foods-10-03056-f001:**
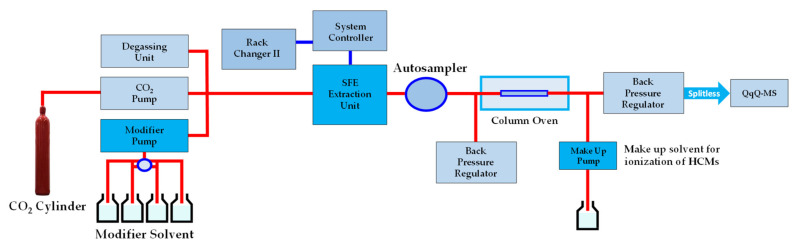
Schematic representation of the online supercritical fluid extraction–supercritical fluid chromatography–mass spectrometry (SFE-SFC-TQ-MS/MS).

**Figure 2 foods-10-03056-f002:**
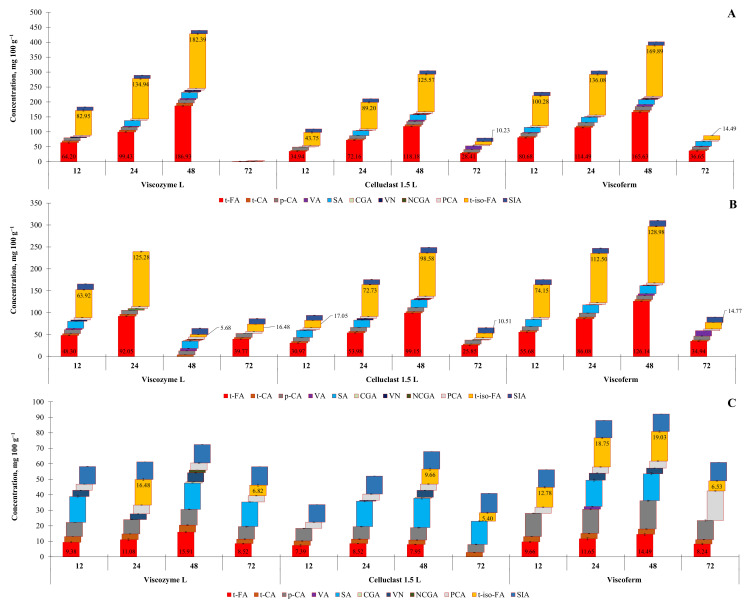
The release of FA and other HCMs using enzymatic hydrolysis of rye (**A**), wheat (**B**), and oat (**C**) bran applying three multi-enzyme complexes, i.e., Viscozyme L, Celluclast 1.5 L, and Viscoferm (mg 100 g^−1^ DW). Note: Values are means ± SD values of triplicates (*n* = 3).

**Figure 3 foods-10-03056-f003:**
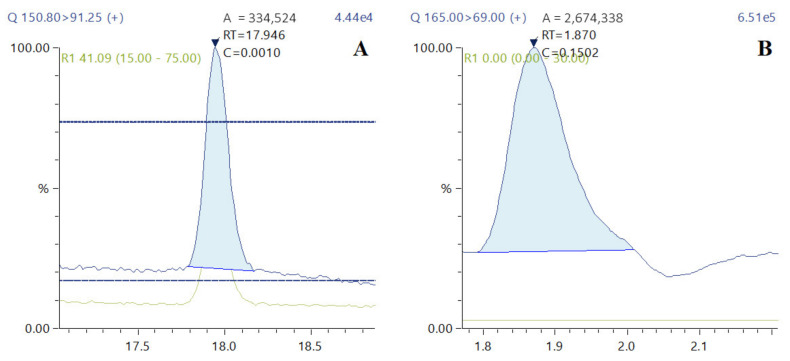
Extracted ion chromatograms (EICs) plotted for the degradation products of 2-methoxy-4-vinylphenol (**A**) and 4-allyl-2-methoxyphenos (**B**).

**Figure 4 foods-10-03056-f004:**
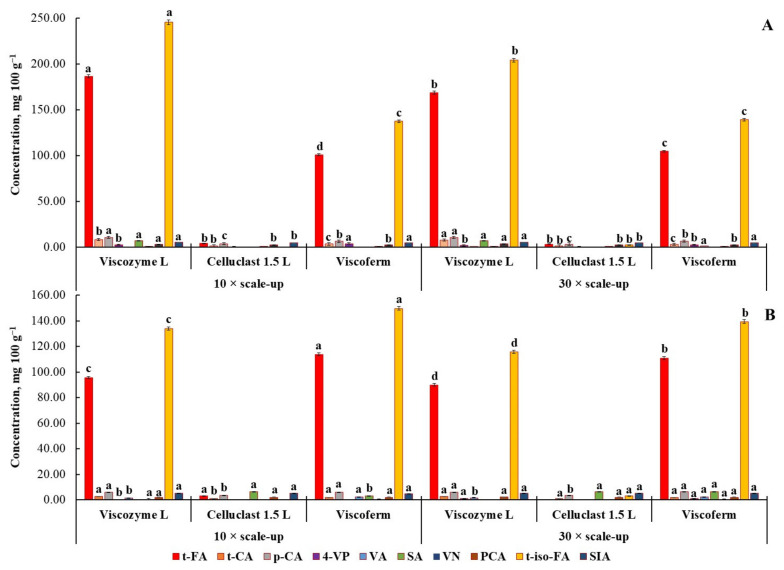
The release of FA and other HCMs after enzymatic hydrolysis of rye (**A**) and wheat (**B**) bran for 48 h with three multi-enzyme complexes, i.e., Viscozyme L, Celluclast 1.5 L, and Viscoferm (mg 100 g^−1^ DW). Note: The process was a 10- and 30-fold scale-up process. Values are means ± SD of triplicates (*n* = 3). Means within the same HCMs with different superscript letters (^a,b,c,d^) are significantly different at *p* < 0.05.

**Figure 5 foods-10-03056-f005:**
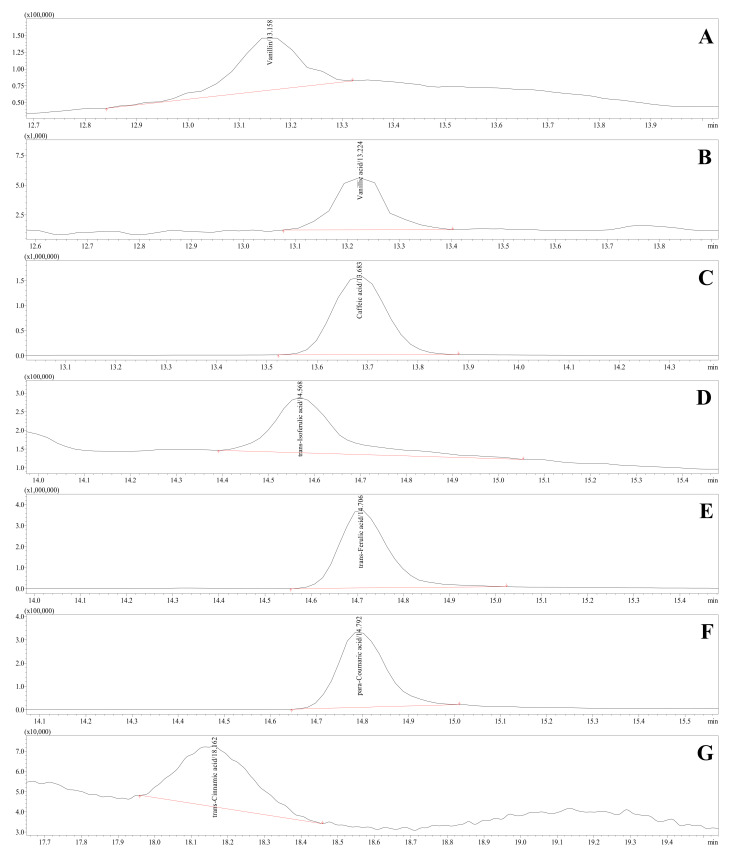
Representative profile of HCMs detected in rye bran hydrolysates, VN (**A**), VA (**B**), CA (**C**), *t*-iso-FA (**D**), *t*-FA (**E**), *p*-CA (**F**), and *t*-CA (**G**), using HPLC-ESI-TQ-MS/MS operating in the multiple reaction monitoring mode.

**Figure 6 foods-10-03056-f006:**
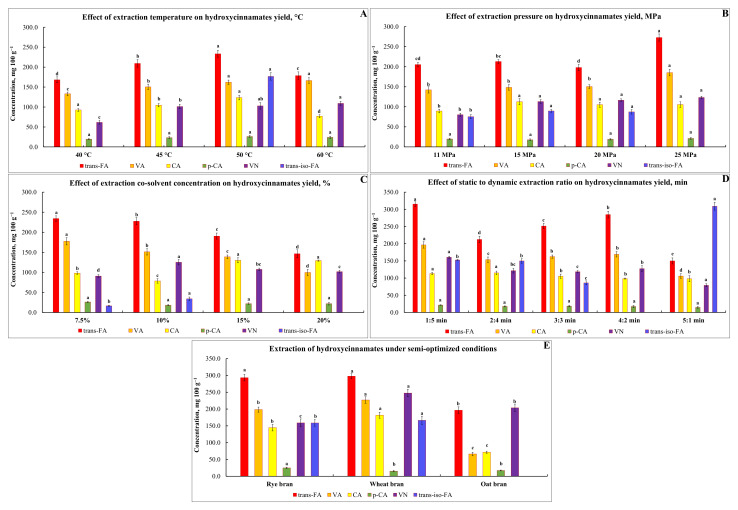
The release, isolation, and analysis of rye-bran-derived HCMs using variable (**A**–**D**) and semi-optimized SFE-CO_2_ extraction conditions (**E**) for extraction of HCMs from bran samples (mg 100 g^−1^ DW). Note: Values are means ± SD of triplicates (*n* = 3). Means within the same HCM with different superscript letters (^a,b,c,d^) are significantly different at *p* < 0.05.

**Figure 7 foods-10-03056-f007:**
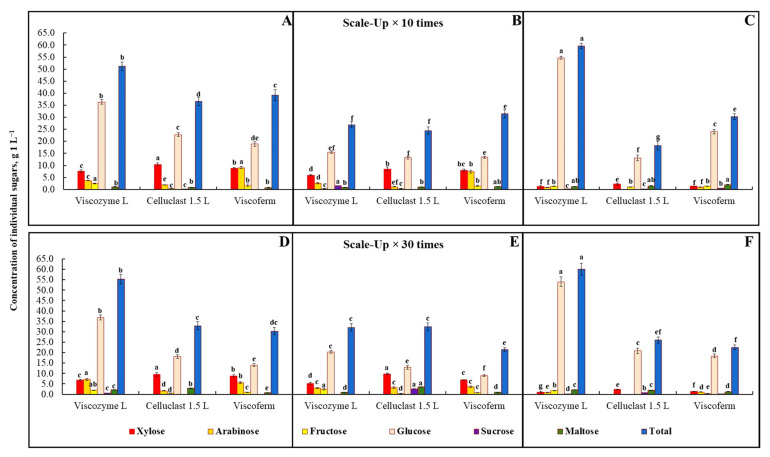
The release of mono- and disaccharides after enzymatic hydrolysis of rye (**A**,**D**), wheat (**B**,**E**), and oat (**C**,**F**) bran samples for 48 h with three multi-enzyme complexes, i.e., Viscozyme L, Celluclast 1.5 L, and Viscoferm (g 1000 mL^−1^ of hydrolysate). Note: The process was performed under 10- and 30-fold scale-up process conditions. Values are means ± SD of triplicates (*n* = 3). Means within the same sugar with different superscript letters (^a,b,c,d,e,f^) are significantly different at *p* < 0.05.

**Figure 8 foods-10-03056-f008:**
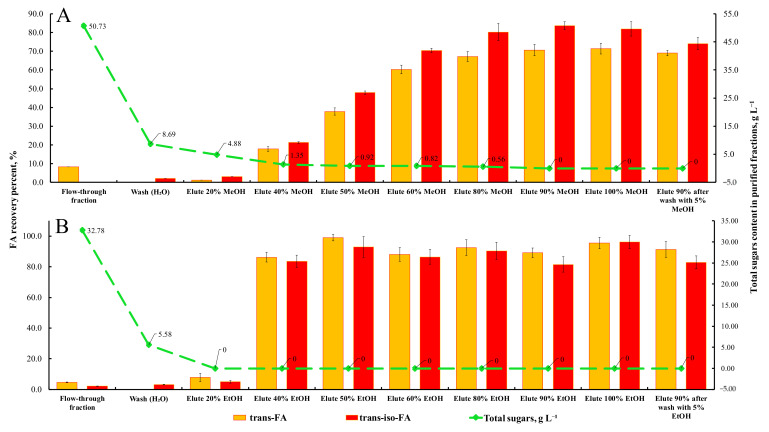
The recovery percentage of *t*-FA and *t*-iso-FA following solid-phase extraction by a Strata X column using MeOH (**A**) and EtOH (**B**) and their aqueous solutions as eluents. Note: Values are means ± SD of triplicates (*n* = 3). The green line on the plot indicates the residual concentration of sugars (g L^−1^) in the fractions collected. −5.00.

**Table 1 foods-10-03056-t001:** Nutritional composition of bran by-products derived from rye, wheat, and oat grains (g 100 g^−1^ DW).

Major Nutrient Profile, g 100 g^−1^ DW								
						FA, mg 100 g^−1^		
Type of Bran	Moisture, %	CHs	Crude Lipids	Crude Proteins	DF	Free	Thermally Processed	Alkali- Hydrolyzed
Rye	11.7± 0.2 ^a^	30.9 ± 0.5 ^b^	3.8 ± 0.1 ^c^	16.9± 0.5 ^a^	36.0 ± 1.9 ^b^	19.6 ± 0.6 ^b^	14.2 ± 0.1 ^b^	171.2 ± 3.2 ^b^
Wheat	11.9± 0.2 ^a^	20.3 ± 0.4 ^c^	4.5 ± 0.1 ^b^	16.2 ± 0.4 ^a^	46.5 ± 2.1 ^a^	32.1 ± 0.7 ^a^	19.9 ± 0.1 ^a^	273.3 ± 4.1 ^a^
Oat	12.4± 0.3 ^a^	50.0 ± 0.9 ^a^	6.7 ± 0.5 ^a^	14.0 ± 0.7 ^b^	14.0 ± 1.7 ^c^	5.7 ± 0.1 ^c^	10.5 ± 0.1 ^c^	45.7. ± 1.9 ^c^

Note: Values are means ± SD values of three replicates (*n* = e). Means within the same column with different superscript letters (^a,b,c^) are significantly different at *p* < 0.05. DW—dry weight; CHs—carbohydrates; DF—dietary fiber. Free, thermally processed, and alkali-hydrolyzed refer to the extraction of FA from bran material based on solid–liquid extraction and alkaline-assisted hydrolysis following the methodology described by Juhnevica-Radenkova et al. [14].

**Table 2 foods-10-03056-t002:** The list of commercial hydrolytic enzymes used in this study.

Commercial Enzyme	Declared Activity	Enzyme Activity	Source	EC Number
Viscoferm^®^	222 FBG g^−1^	Endo-1,4-β-xylanase, endo-1,3-(1,4)-β-d-glucanase	*Aspergillus* spp.	3.2.1.8 3.2.1.4
Viscozyme^®^ L	100 FBG g^−1^	Endo-1,4-β-xylanase, non-reducing end α-L-arabinofuranosidase, endo-1,4-β-d-glucanase	*Aspergillus* *aculeatus*	3.2.1.8 3.2.1.55 3.2.1.4
Celluclast^®^ 1.5 L	700 EGU g^−1^	Endo-1,4-β-d-glucanase	*Trichoderma reesei*	3.2.1.4

Note: EC—enzyme commission; EGU—endoglucanase units; FBG—fungal β-glucanase units.

**Table 3 foods-10-03056-t003:** The experimental layout uses an orthogonal array design.

Experiment No.	Temperature, °C	Pressure, MPa	Co-Solvent Concentration, %	Extrication Time (Static to Dynamic Mode), min
1	40	15	10	3:3
2	45	15	10	3:3
3	50	15	10	3:3
4	60	15	10	3:3
5	50	10	10	3:3
6	50	15	10	3:3
7	50	20	10	3:3
8	50	25	10	3:3
9	50	15	7.5	3:3
10	50	15	10	3:3
11	50	15	15	3:3
12	50	15	20	3:3
13	50	15	10	1:5
14	50	15	10	2:4
15	50	15	10	3:3
16	50	15	10	4:2
17	50	15	10	5:1

**Table 4 foods-10-03056-t004:** MRM transitions, collision energy, Q1, Q3, and dwell time for investigated hydroxycinnamates.

Compound	Retention Time, min	Molecular Formula	Ionization Mode	MRM Transitions	Q1 Pre-Bias, V	Collision Energy, V	Q3 Pre-Bias, V	Dwell Time, msec
Eugenol	1.953	C_10_H_12_O_2_	[M + H]^+^	165.0000→69.0000	−11.0	−22.0	−15.0	20.0
				165.0000→109.0000	−11.0	−24.0	−13.0	20.0
				165.0000→137.0500	−11.0	−13.0	−16.0	20.0
Gallic acid	9.033	C_7_H_6_O_5_	[M − H]^−^	169.0000→124.9000	12.0	17.0	10.0	20.0
				169.0000→78.9500	12.0	24.0	15.0	20.0
				169.0000→68.9000	12.0	22.0	11.0	20.0
Neochlorogenic acid	10.245	C_16_H_18_O_9_	[M − H]^−^	353.1000→191.0500	13.0	22.0	20.0	20.0
				353.1000→135.0000	13.0	31.0	12.0	20.0
				353.1000→179.0500	13.0	19.0	10.0	20.0
Protocatechuic acid	11.515	C_7_H_6_O_4_	[M − H]^−^	153.2000→108.9500	10.0	16.0	20.0	20.0
				153.2000→107.9500	10.0	24.0	22.0	20.0
				153.2000→91.0500	10.0	26.0	17.0	20.0
Chlorogenic acid	12.044	C_16_H_18_O_9_	[M − H]^−^	353.1000→191.1000	19.0	22.0	20.0	20.0
				353.1000→85.0500	13.0	43.0	16.0	20.0
				353.1000→127.0000	13.0	36.0	10.0	20.0
(+)−Catechin	12.703	C_15_H_14_O_6_	[M − H]^−^	288.9500→245.0000	14.0	15.0	14.0	20.0
				288.9500→109.0000	14.0	26.0	19.0	20.0
				288.9500→123.0000	14.0	31.0	10.0	20.0
(−)−Epicatechin	12.709	C_15_H_14_O_6_	[M − H]^−^	289.0500→245.0000	14.0	16.0	14.0	20.0
				289.0500→109.0000	14.0	26.0	20.0	20.0
				289.0500→123.0000	14.0	30.0	22.0	20.0
Syringic acid	13.033	C_9_H_10_O_5_	[M − H]^−^	197.1500→182.1000	20.0	15.0	10.0	20.0
				197.1500→122.9500	17.0	24.0	19.0	20.0
				197.1500→95.1000	12.0	31.0	18.0	20.0
Vanillic acid	13.248	C_8_H_8_O_4_	[M − H]^−^	167.0500→152.0000	12.0	18.0	30.0	20.0
				167.0500→107.9000	12.0	19.0	20.0	20.0
				167.0500→123.0000	12.0	14.0	19.0	20.0
Vanillin	13.366	C_8_H_8_O_3_	[M + H]^+^	152.9500→65.1000	−10.0	−24.0	−24.0	20.0
				152.9500→93.0500	−10.0	−16.0	−20.0	20.0
				152.9500→125.1000	−10.0	−15.0	−15.0	20.0
4-Vinylphenol	13.747	C_8_H_8_O	[M + H]^+^	121.0500→77.2000	−20.0	−23.0	−16.0	20.0
				121.0500→91.2000	−21.0	−19.0	−18.0	20.0
				121.0500→51.2000	−22.0	−36.0	−21.0	20.0
Caffeic acid	13.714	C_9_H_8_O_4_	[M − H]^−^	179.1500→135.0000	12.0	18.0	25.0	20.0
				179.1500→134.0000	12.0	25.0	24.0	20.0
Sinapic acid	14.361	C_11_H_12_O_5_	[M + H]^+^	224.9000→207.1000	−15.0	−10.0	−16.0	20.0
				224.9000→161.0000	−10.0	−10.0	−18.0	20.0
				224.9000→91.1500	−15.0	−27.0	−21.0	20.0
*trans*-Isoferulic acid	14.728	C_10_H_10_O_4_	[M + H]^+^	194.9000→177.1000	−13.0	−11.0	−13.0	20.0
				194.9000→131.0000	−13.0	−11.0	−15.0	20.0
				194.9000→89.1500	−13.0	−32.0	−19.0	20.0
*trans*-Ferulic acid	14.733	C_10_H_10_O_4_	[M − H]^−^	193.0500→134.0000	10.0	18.0	23.0	20.0
				193.0500→178.0500	10.0	15.0	15.0	20.0
*para*-Coumaric acid	14.818	C_9_H_8_O_3_	[M − H]^−^	163.0500→119.0500	11.0	16.0	21.0	20.0
				163.0500→93.0500	12.0	31.0	17.0	20.0
2-Methoxy-4- vinylphenol	17.972	C_9_H_10_O_2_	[M + H]^+^	150.8000→91.2500	−10.0	−22.0	−19.0	20.0
				150.8000→65.2000	−25.0	−32.0	−28.0	20.0
*trans*−Cinnamic acid	18.600	C_9_H_8_O_2_	[M + NH4]^+^	165.0000→101.2000	−12.0	−12.0	−20.0	20.0
				165.0000→69.2000	−12.0	−23.0	−29.0	20.0
				165.0000→133.2500	−11.0	−10.0	−15.0	20.0

The first MRM transitions found were used for qualitative analysis and the second or third for quantitative analysis.

**Table 5 foods-10-03056-t005:** The release kinetics of mono- and disaccharides during 72 h enzymatic hydrolysis of bran samples using three multi-enzyme complexes, i.e., Viscozyme L, Celluclast 1.5 L, and Viscoferm (g 1000 mL^−1^ of bran hydrolysate).

Enzyme	Viscozyme L				Celluclast 1.5 L				Viscoferm			
Time, h	12	24	48	72	12	24	48	72	12	24	48	72
Carbohydrate Xyl					Rye Bran							
	6.8 ± 0.6 ^c^	9.4 ± 0.5 ^b^	12.9 ± 0.0 ^a^	9.7 ± 0.0 ^b^	10.6 ± 0.6 ^b^	12.3 ± 1.0 ^a^	12.8 ± 0.6 ^a^	11.1 ± 0.4 ^b^	8.9 ± 0.5 ^c^	9.9 ± 0.5 ^b^	10.8 ± 0.2 ^a^	10.9 ± 0.6 ^a^
Ara	5.3 ± 0.5 ^a^	4.2 ± 0.5 ^b^	5.1 ± 0.0 ^a^	3.2 ± 0.0 ^c^	2.5 ± 0.1 ^a^	2.7 ± 0.3 ^a^	2.5 ± 0.2 ^a^	2.2 ± 0.2 ^a^	8.0 ± 0.1 ^a^	7.4 ± 1.0 ^a^	4.6 ± 0.4 ^c^	5.7 ± 0.4 ^b^
Fru	2.5 ± 0.3 ^c^	3.0 ± 0.3 ^b^	4.4 ± 0.1 ^a^	1.6 ± 0.0 ^d^	2.6 ± 0.1 ^a^	2.9 ± 0.0 ^a^	3.1 ± 0.1 ^a^	2.5 ± 0.2 ^a^	2.2 ± 0.2 ^b^	3.1 ± 0.4 ^a^	2.2 ± 0.1 ^b^	2.9 ± 0.1 ^a b^
Glu	22.2 ± 0.4 ^d^	34.1 ± 0.9 ^b^	39.9 ± 0.9 ^a^	31.5 ± 1.0 ^c^	11.5 ± 0.2 ^b^	13.4 ± 0.8 ^a^	13.3 ± 0.9 ^a^	11.8 ± 0.6 ^b^	10.8 ± 0.4 ^b^	10.8 ± 0.8 ^b^	10.7 ± 0.1 ^b^	19.9 ± 0.7 ^a^
Suc	0.2 ± 0.0	0.0 ± 0.0	0.0 ± 0.0	0.0 ± 0.0	0.0 ± 0.0	0.0 ± 0.0	0.0 ± 0.0	0.0 ± 0.0	0.4 ± 0.1	0.0 ± 0.0	0.0 ± 0.0	0.0 ± 0.0
Mal	0.8 ± 0.1 ^b^	1.4 ± 0.0 ^a^	1.3 ± 0.0 ^a b^	1.4 ± 0.0 ^a^	0.6 ± 0.0 ^a^	1.3 ± 0.1 ^a^	1.4 ± 0.1 ^a^	1.5 ± 0.0 ^a^	0.5 ± 0.1 ^b^	1.8 ± 0.1 ^a^	1.7 ± 0.0 ^a^	1.7 ± 0.1 ^a^
Total	37.8 ± 1.9 ^d^	52.1 ± 2.2 ^b^	63.6 ± 1.02 ^a^	47.4 ± 1.0 ^c^	27.8 ± 1.0 ^b^	32.6 ± 2.2 ^a^	33.1 ± 1.7 ^a^	29.1 ± 1.4 ^a b^	30.8 ± 1.4 ^c^	33.0 ± 2.8 ^b^	30.0 ± 0.8 ^c^	41.1 ± 1.9 ^a^
Carbohydrate Xyl					Wheat Bran							
	6.4 ± 0.6 ^c^	8.2 ± 0.1 ^b^	10.6 ± 0.4 ^a^	8.9 ± 0.6 ^b^	8.6 ± 0.4 ^d^	10.7 ± 0.2 ^b^	12.1 ± 0.2 ^a^	9.5 ± 0.2 ^c^	1.4 ± 0.1 ^c^	8.6 ± 0.4 ^b^	9.7 ± 0.5 ^a^	10.3 ± 0.5 ^a^
Ara	1.9 ± 0.1 ^c^	3.1 ± 0.2 ^b^	5.1 ± 0.2 ^a^	2.8 ± 0.3 ^b^	2.1 ± 0.2 ^b^	3.6 ± 0.1 ^a^	4.1 ± 0.1 ^a^	2.0 ± 0.1 ^b^	1.2 ± 0.1 ^c^	4.0 ± 0.4 ^b^	4.3 ± 0.2 ^a b^	5.0 ± 0.2 ^a^
Fru	1.0 ± 0.1 ^c^	1.9 ± 0.1 ^b^	4.0 ± 0.1 ^a^	1.5 ± 0.2 ^b c^	2.5 ± 0.1 ^b^	3.4 ± 0.2 ^a^	3.7 ± 0.2 ^a^	2.6 ± 0.1 ^b^	0.9 ± 0.0 ^c^	1.8 ± 0.1 ^b^	2.5 ± 0.1 ^a b^	2.9 ± 0.4 ^a^
Glu	13.8 ± 0.4 ^c^	19.0 ± 0.8 ^b^	25.1 ± 0.1 ^a^	18.7 ± 0.3 ^b^	5.9 ± 0.1 ^c^	6.2 ± 0.2 ^b c^	6.7 ± 0.3 ^a b^	7.4 ± 0.3 ^a^	17.3 ± 0.4 ^a^	6.6 ± 0.1 ^c^	7.4 ± 0.1 ^c^	16.0 ± 0.7 ^b^
Suc	1.2 ± 0.1 ^a^	0.6 ± 0.1 ^a^	0.4 ± 0.0 ^a^	0.1 ± 0.0 ^b^	0.5 ± 0.1	0.0 ± 0.0	0.0 ± 0.0	0.0 ± 0.0	0.0 ± 0.0	0.0 ± 0.0	0.0 ± 0.0	0.0 ± 0.0
Mal	1.0 ± 0.1 ^a^	1.6 ± 0.1 ^a^	1.6 ± 0.1 ^a^	1.9 ± 0.0 ^a^	1.1 ± 0.1 ^a^	1.1 ± 0.0 ^a^	1.3 ± 0.0 ^a^	1.3 ± 0.1 ^a^	0.4 ± 0.1 ^b^	1.3 ± 0.1 ^a^	1.4 ± 0.1 ^a^	1.6 ± 0.1 ^a^
Total	25.3 ± 1.4 ^c^	34.4 ± 1.4 ^b^	46.8 ± 0.9 ^a^	33.9 ± 1.4 ^b^	20.7 ± 1.0 ^d^	25.0 ± 0.7 ^b^	27.9 ± 0.8 ^a^	22.8 ± 0.8 ^c^	21.2 ± 0.7 ^d^	22.3 ± 1.1 ^c^	25.3 ± 1.0 ^b^	35.8 ± 1.9 ^a^
Carbohydrate Xyl					Oat Bran							
	1.2 ± 0.1 ^a^	1.4 ± 0.1 ^a^	1.6 ± 0.1 ^a^	1.9 ± 0.1 ^a^	1.6 ± 0.1 ^b^	3.1 ± 0.1 ^a^	1.5 ± 0.1 ^b^	1.9 ± 0.0 ^b^	1.4 ± 0.0 ^a^	1.4 ± 0.0 ^a^	1.6 ± 0.1 ^a^	1.8 ± 0.1 ^a^
Ara	1.1 ± 0.1 ^a^	1.2 ± 0.1 ^a^	1.1 ± 0.1 ^a^	1.6 ± 0.1 ^a^	0.0 ± 0.1	0.0 ± 0.0	0.0 ± 0.0	0.0 ± 0.0	1.3 ± 0.1 ^a^	1.1 ± 0.1 ^a^	1.1 ± 0.1 ^a^	1.3 ± 0.0 ^a^
Fru	0.0 ± 0.0	0.0 ± 0.0	0.0 ± 0.0	0.0 ± 0.0	0.0 ± 0.0	0.0 ± 0.0	0.0 ± 0.0	0.0 ± 0.0	0.7 ± 0.0 ^a^	0.8 ± 0.1 ^a^	0.9 ± 0.1 ^a^	1.0 ± 0.1 ^a^
Glu	56.6 ± 1.9 ^c^	62.3 ± 1.6 ^a^	59.5 ± 1.2 ^b^	55.5 ± 2.1 ^c^	7.0 ± 0.3 ^d^	10.4 ± 0.6 ^c^	12.6 ± 0.4 ^b^	17.1 ± 0.4 ^a^	13.9 ± 0.6 ^d^	16.2 ± 0.9 ^c^	20.3 ± 0.1 ^b^	22.6 ± 0.5 ^a^
Suc	0.0 ± 0.0	0.4 ± 0.1	0.0 ± 0.0	0.0 ± 0.0	0.0 ± 0.0 ^b^	1.1 ± 0.1 ^a^	1.4 ± 0.1 ^a^	1.6 ± 0.1 ^a^	0.0 ± 0.0	0.0 ± 0.0	0.0 ± 0.0	0.0 ± 0.0
Mal	0.6 ± 0.1 ^a^	1.0 ± 0.0 ^a^	1.0 ± 0.1 ^a^	1.0 ± 0.1 ^a^	0.3 ± 0.1 ^a^	0.5 ± 0.0 ^a^	0.5 ± 0.1 ^a^	0.6 ± 0.1 ^a^	0.4 ± 0.1 ^b^	1.5 ± 0.1 ^a^	2.2 ± 0.1 ^a^	2.1 ± 0.1 ^a^
Total	59.5 ± 2.2 ^c^	66.3 ± 1.9 ^a^	63.2 ± 1.5 ^b^	60.0 ± 2.4 ^c^	8.9 ± 0.6 ^d^	15.1 ± 0.8 ^c^	16.0 ± 0.7 ^b^	21.2 ± 0.6 ^a^	17.7 ± 0.8 ^d^	21.0 ± 1.2 ^c^	26.1 ± 0.5 ^b^	28.8 ± 0.91 ^a^

Note: Values are means ± SD of three replicates (*n* = 3). Xyl—xylose; Ara—arabinose; Fru—fructose; Glu—glucose; Suc—sucrose; Mal—maltose. Means within the same carbohydrate with different superscript letters (^a,b,c,d^) are significantly different at *p* < 0.05.

**Table 6 foods-10-03056-t006:** Fatty acid composition of bran hydrolysates according to the hydrolysis method.

Type of Bran	HM	Fatty Acid, % *w/v*							
		SFA		MUFA			PUFA		
		C16:0	C18:0	C18:1n9c	C18:1n9t	C20:1n9c	C18:2n6c	C18:3n3c	C20:3n3c
Rye	KOH (10% in MeOH)	5.46 ± 0.06	0.33 ± 0.00	4.18 ± 0.19	nd	0.06 ± 0.01	14.92 ± 0.13	1.27 ± 0.04	<LOQ
	Viscozyme L	14.24 ± 1.16	0.89 ± 0.13	12.13 ± 1.36	nd	2.46 ± 0.74	36.10 ± 1.24	2.62 ± 0.26	nd
	Celluclast 1.5 L	7.97 ± 0.61	0.37 ± 0.09	5.88 ± 0.44	nd	nd	22.72 ± 0.26	1.58 ± 0.10	nd
	Viscoferm	4.39 ± 0.10	0.31 ± 0.02	3.22 ± 0.34	nd	nd	12.92 ± 0.36	0.59 ± 0.20	nd
Wheat	KOH (10% in MeOH)	5.62 ± 0.08	0.38 ± 0.02	5.40 ± 0.08	nd	0.01 ± 0.0 0	14.80 ± 0.68	1.10 ± 0.07	0.02 ± 0.01
	Viscozyme L	10.19 ± 0.29	0.67 ± 0.06	10.86 ± 0.91	0.05 ± 0.02	0.23 ± 0.05	27.00 ± 0.99	2.34 ± 0.14	nd
	Celluclast 1.5 L	22.88 ± 0.39	1.22 ± 0.01	21.06 ± 0.31	nd	nd	57.09 ± 0.42	4.34 ± 0.06	nd
	Viscoferm	6.80 ± 0.30	0.45 ± 0.12	6.64 ± 0.30	nd	nd	17.87 ± 0.40	0.72 ± 0.03	nd
Oat	KOH (10% in MeOH)	7.12 ± 0.05	0.54 ± 0.01	13.15 ± 0.66	nd	0.12 ± 0.01	15.30 ± 0.35	0.54 ± 0.10	nd
	Viscozyme L	12.37 ± 1.35	1.01 ± 0.08	29.02 ± 3.75	nd	0.43 ± 0.05	28.14 ± 4.49	1.10 ± 0.19	nd
	Celluclast 1.5 L	12.20 ± 0.25	0.75 ± 0.15	27.88 ± 0.27	nd	nd	25.89 ± 0.70	0.66 ± 0.28	nd
	Viscoferm	8.32 ± 0.01	0.66 ± 0.08	18.62 ± 0.31	nd	nd	18.35 ± 0.38	0.19 ± 0.10	nd

Note: HM—hydrolysis method; SFA—saturated fatty acids; MUFA—monounsaturated fatty acids; PUFA—polyunsaturated fatty acids; nd—not detected; <LOQ—below limit of quantification. Values are means ± SD of triplicates (*n* = 3).

## Data Availability

The data sets and analysis of the study are available from the corresponding author upon reasonable request.

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
