# Peer review of "Lignocellulose-Degrading Enzymes: A Biotechnology Platform for Ferulic Acid Production from Agro-Industrial Side Streams"

_foods, 2021, doi:10.3390/foods10123056_

Round 1

Reviewer 1 Report

The present manuscript is dealing with the extraction of ferulic acid from lignocellulosic food by-products, through enzymatic assisted processes. The topic is interesting and the work well designed and performed. The manuscript meets the quality standards in Foods.

Minor corrections:

P1 L27: 100 g−1 (please, put -1 as superscript character)

P1 L30: 100 g−1 (please, put -1 as superscript character)

P2 L51-L54: please, rephrase

P7 L250: Supercritical fluid extraction-supercritical fluid chromatography-mass spectrometry (SFE- SFC-MS-TQ)

P10 L330: paragraphs 2.14; 2.15 and 2.16 can be combined into a single paragraph

P11 L368-L389: It is redundant, as it repeats concepts already highlighted in the introduction section.

P13 L436-443: it is superfluous.

Table 5: the table is difficult to understand. Furthermore, statistical significance is missing.

P25 L843: “the validity of the technology”. What technology?

P25 L845: Since the yield with SFE- SFC-MS-TQ appears to be higher than enzymatic hydrolysis, authors should state within the Conclusion section why enzymatic hydrolysis should or could be preferred.

Author Response

Dear Reviewer,

R: The present manuscript is dealing with the extraction of ferulic acid from lignocellulosic food by-products, through enzymatic assisted processes. The topic is interesting and the work well designed and performed. The manuscript meets the quality standards in Foods.

A: We would like to thank for careful and thorough checking of our manuscript and valuable comments. In preparing the manuscript authors have incorporated most of the changes suggested. The authors refer to them in detail below.

Minor corrections:

R: P1 L27: 100 g−1 (please, put -1 as superscript character)

A: The authors thank the reviewer for his valuable observation. 

R: P1 L30: 100 g−1 (please, put -1 as superscript character)

A: The authors thank the reviewer for his valuable observation. 

R: P2 L51-L54: please, rephrase

A: The following fragment has been rephrased: “The presence of complex dietary fiber in the matrix of bran makes this material suitable for livestock feeding, while, their application in the food industry is negligible.”

R: P7 L250: Supercritical fluid extraction-supercritical fluid chromatography-mass spectrometry (SFE- SFC-MS-TQ)

A: The authors thank the reviewer for his valuable observation. 

R: P10 L330: paragraphs 2.14; 2.15 and 2.16 can be combined into a single paragraph

A: Dear reviewer! The authors decided to split this information into three parts to make this easier for perception and to not confuse the readers who are not specialists in chromatography by difficult technical terms. The authors try to adhere to the same style in all their works, and therefore the authors wish to leave this fragment in the present style.

R: P11 L368-L389: It is redundant, as it repeats concepts already highlighted in the introduction section.

A: The authors consent with the reviewer that the following fragment represents some similarity with the information provided in the Introduction section. We have revised this fragment leaving just a few sentences in the paragraph.

R: P13 L436-443: it is superfluous.

A: Dear reviewer! The main intention of this fragment was to highlight that literature still represents some limitations regarding the analysis of iso-ferulic acid in other plant matrices. Despite the fact that this molecule has been well-documented in such plant material as Cimicifugae rhizoma, there are still some obscure points in its exact concentrations. The authors wish to highlight that the grain-derived hydrolysates might represent potential interest as raw material for isolation of iso-ferulic acid.

R: Table 5: the table is difficult to understand. Furthermore, statistical significance is missing.

A: The authors thank the reviewer for fair remark. We have amended this table and hope now it appears in a good shape.

R: P25 L843: “the validity of the technology”. What technology?

A: The authors meant technology of enzymatic hydrolysis. This fragment has been corrected:

R: P25 L845: Since the yield with SFE- SFC-MS-TQ appears to be higher than enzymatic hydrolysis, authors should state within the Conclusion section why enzymatic hydrolysis should or could be preferred.

A: The authors appreciate the valuable remark of the reviewer. Additional statement with this regard has been provided within the conclusion section: “Compared to the SFE technique, the EH could be considered an affordable alternative, which can represent the future of sustainable FA production. While side products mono- and disaccharides could be exploited as renewable raw material for bioethanol fuel production or along with remained solids might be introduced in the feed formulations for cattle. In the context of non-waste technology, a proposed method has the potential to be used under large-scale conditions, where industrial symbiosis by combining the grain milling facility as a supplier of bran and biorefinery for the production of FA can be demonstrated.

The authors hope that the above explanations and adherence to the suggestions made in the reviews will render the attached manuscript appropriate and free from any understatement.

Reviewer 2 Report

The manuscript is well written and all the important information is included. However, I have some concerns which need to be addressed 
1) It would be interesting to include to the TIC and XIC for the phenolics by HPLC-ESI-QqQ-MS
2) The current article is already published and available:
 https://www.preprints.org/manuscript/202111.0130/v1

Is it advisable to publish it again? 

Author Response

Dear Reviewer,

R: The manuscript is well written and all the important information is included. However, I have some concerns which need to be addressed

A: We would like to thank for careful and thorough checking of our manuscript and valuable comments. In preparing the manuscript authors have incorporated most of the changes suggested. The authors refer to them in detail below.

R: 1) It would be interesting to include to the TIC and XIC for the phenolics by HPLC-ESI-QqQ-MS

A: Dear reviewer. Since this study deals with the analysis of ferulic acid and other hydroxycinnamates applying a selective and more accurate approach based on MRM transitions, additional total ion chromatogram in the authors’ view will bring neither informative nor pithy value to the manuscript.

Some of the EICs (XICs) have already been plotted for the degradation products of 2-methoxy-4-vinylphenol (A) and 4-allyl-2-methoxyphenos (B) (Figure 3) and HCMs (Figure 5) in the manuscript.

R: 2) The current article is already published and available:

 https://www.preprints.org/manuscript/202111.0130/v1

Is it advisable to publish it again?

A: Dear reviewer! The authors understand your concern regarding already published version of the manuscript “Lignocellulose-Degrading Enzymes: A Biotechnology Platform for Ferulic Acid Production from Agro-Industrial Side Streams” which is available from the Preprints Platform However, the authors want to inform the reviewer that the version of a scientific manuscript posted on this platform could not be treated as a publication but rather as an early version of research outputs which has not undergone formal peer review. This article appears exclusively as online version and is freely available.

This service is provided along with the submission of the manuscript to any MDPI series journals. 

The authors hope that the above explanations and adherence to the suggestions made in the reviews will render the attached manuscript appropriate and free from any understatement.

Reviewer 3 Report

The authors present a work entitled Lignocellulose-Degrading Enzymes: A Biotechnology Platform for Ferulic Acid Production from Agro-Industrial Side Streams ". The manuscript is well written and organized, it presents important innovations both from the point of view of the eco-friendly recovery of substances from 'high added value starting from poor sources, both from an analytical point of view with the coupling of an extraction system to one of structural characterization. In order to improve the general quality of the work, I suggest only some minor changes:

In the abstract and throughout the text, explain the acronyms the first time they appear.

Generally, the authors should pay more attention to the expressions of the English language, several times they use sentences that are too long and difficult to understand

Ln 51-54: “The presence of cellu- 51 lose, hemicellulose, and lignin, the way the bioactive compounds are integrated into the matrix, along with adverse effects on technological processes, the application of bran in the food industry plays a secondary role.” Please, explain this statement more clearly

Ln 59: please, replace synthesizes by synthesis

Ln 530-535: “Due to the GRAS status assigned and relatively low costs, CO2 is the most widely used supercritical fluid suitable for both research purposes and industrial scales. Since neat supercritical fluid CO2 has dissolving propertied close to hexane that is recognized as an excellent solvent for extracting non-polar compounds, the addition of co-solvent could enhance the solubilizing properties of CO2, making it attainable to recover more polar molecules.” This is a very important assumption, however the authors should highlight that the extraction with supercritical CO2 is not a low cost technique and its use must always be justified by the possible added value of the extractable substances. In this regard, the authors should cite literature and explain / justify the use of this technique and the advantages compared to projects in the literature that use other low-cost techniques.

In the “Conclusions” the authors should briefly illustrate some concrete application of the proposed method

Author Response

Dear Reviewer,

R: The authors present a work entitled Lignocellulose-Degrading Enzymes: A Biotechnology Platform for Ferulic Acid Production from Agro-Industrial Side Streams ". The manuscript is well written and organized, it presents important innovations both from the point of view of the eco-friendly recovery of substances from 'high added value starting from poor sources, both from an analytical point of view with the coupling of an extraction system to one of structural characterization. In order to improve the general quality of the work, I suggest only some minor changes:

A: We would like to thank for careful and thorough checking of our manuscript and valuable comments. In preparing the manuscript authors have incorporated most of the changes suggested. The authors refer to them in detail below.

R: In the abstract and throughout the text, explain the acronyms the first time they appear.

A:  The authors thank the reviewer for the valuable remark. Now, each acronym the first time it appears in the manuscript is explained.

R: Generally, the authors should pay more attention to the expressions of the English language, several times they use sentences that are too long and difficult to understand

A: The authors have proofread the manuscript, and hope that now the English quality is much better.

R: Ln 51-54: “The presence of cellu- 51 lose, hemicellulose, and lignin, the way the bioactive compounds are integrated into the matrix, along with adverse effects on technological processes, the application of bran in the food industry plays a secondary role.” Please, explain this statement more clearly

A: Dear reviewer! The authors made a change to this fragment to get it ieaser to understand. Now it apprears as follows:  “The presence of indigestible dietary fiber in the matrix of bran makes this material suitable for livestock feeding, while, their application in the food industry is negligible.”

R: Ln 59: please, replace synthesizes by synthesis

A: The authors substitued the word “synthesizes” with “synthesis”

R: Ln 530-535: “Due to the GRAS status assigned and relatively low costs, CO2 is the most widely used supercritical fluid suitable for both research purposes and industrial scales. Since neat supercritical fluid CO2 has dissolving propertied close to hexane that is recognized as an excellent solvent for extracting non-polar compounds, the addition of co-solvent could enhance the solubilizing properties of CO2, making it attainable to recover more polar molecules.” This is a very important assumption, however the authors should highlight that the extraction with supercritical CO2 is not a low cost technique and its use must always be justified by the possible added value of the extractable substances. In this regard, the authors should cite literature and explain / justify the use of this technique and the advantages compared to projects in the literature that use other low-cost techniques.

A: Dear reviewer. The authors have provided additional reference supporting relativelly high consts of SFE. “It is worth noting that technologies involving elevated pressures require high investment costs for high-pressure equipment and therefore the application of SFE must be justified by the extraction of substances which along with added value also highly demanded on the market and are of potential industrial applications [Hrncic et al. 2018].”

 Hrncic, M. K.; Cör, D.; Verboten, M. T.; Knez, Z. Application of supercritical and subcritical fluids in food processing. Food Qual. Saf. 2018, 2, 59–67, doi:10.1093/fqsafe/fyy008.

Besides, one of the main advantage of SFE technique over other extraction approaches was already mentioned in the manuscript: “Due to the process taking place in a closed-loop, the online extraction, separation, and analysis of HCMs using SFE-CO2-SFC coupled to TQ-MS/MS make it attainable to reduce both qualitative and quantitative losses of analytes during analytical work. Given this circumstance, along with reduced solvents consumption, the SFE technique has a significant advantage over the other sample preparation methods [41].

Abbas, K. A.; Mohamed, A.; Abdulamir, A. S.; Abas, H. A. A review on supercritical fluid extraction as new analytical method. Am. J. Biochem. Biotechnol. 2008, 4, 345–353, doi:10.3844/ajbbsp.2008.345.353.

R: In the “Conclusions” the authors should briefly illustrate some concrete application of the proposed method

A: Dear reviewer. The authors appreciate the valuable suggestion. A potential application of the developed method was mentioned in the Conclusion section.

The authors hope that the above explanations and adherence to the suggestions made in the reviews will render the attached manuscript appropriate and free from any understatement.

Round 2

Reviewer 2 Report

Accept in the current form.